# Bioactive Nutritional Components Within the Planetary Health Diet for Preventing Sarcopenic Obesity and Diabetic Sarcopenia: A Systematic Review

**DOI:** 10.3390/nu17233656

**Published:** 2025-11-22

**Authors:** Lia Elvina, Chiao-Ming Chen, Dang Hien Ngan Nguyen, Chun-Che Wei, Chien-Tien Su, Te-Chao Fang, Fandi Sutanto, Sing-Chung Li

**Affiliations:** 1School of Nutrition and Health Sciences, College of Nutrition, Taipei Medical University, Taipei 11031, Taiwan; liaelvina876@gmail.com (L.E.); ga56114001@tmu.edu.tw (C.-C.W.); 2Department of Pharmacy, Institut Bio Scientia Internasional Indonesia, Jakarta 13210, Indonesia; fandi.sutanto@i3l.ac.id; 3Department of Food Science, Nutrition, and Nutraceutical Biotechnology, Shih Chien University, Taipei 10462, Taiwan; charming@g2.usc.edu.tw; 4Department of Nutrition and Food Safety, Faculty of Public Health, Can Tho University of Medicine and Pharmacy, Can Tho City 900000, Vietnam; ndhngan@ctump.edu.vn; 5Department of Family Medicine, Taipei Medical University Hospital, Taipei 11031, Taiwan; ctsu@tmu.edu.tw; 6School of Public Health, College of Public Health, Taipei Medical University, Taipei 11031, Taiwan; 7Division of Nephrology, Department of Internal Medicine, School of Medicine, College of Medicine, Taipei Medical University, Taipei 11031, Taiwan; fangtc@tmu.edu.tw; 8Division of Nephrology, Department of Internal Medicine, Taipei Medical University Hospital, Taipei Medical University, Taipei 11031, Taiwan; 9Taipei Medical University-Research Center of Urology and Kidney, Taipei Medical University, Taipei 11031, Taiwan

**Keywords:** planetary health diet, sarcopenic obesity, diabetic sarcopenia, bioactive nutrients, AMPK–SIRT1–PGC-1α, Akt–mTOR signaling

## Abstract

Background: Sarcopenic obesity (SO) and diabetic sarcopenia (DS) represent overlapping metabolic–musculoskeletal disorders characterized by the coexistence of excessive adiposity, insulin resistance, and progressive muscle wasting. The Planetary Health Diet (PHD), proposed by the EAT–Lancet Commission, emphasizes plant-forward, nutrient-dense, and environmentally sustainable food patterns that may concurrently address metabolic and muscle health. This review aimed to systematically evaluate dietary and bioactive nutritional interventions aligned with the PHD and their effects on muscle mass, strength, metabolism, and underlying mechanisms in SO and DS. Methods: Following PRISMA guidelines, studies published between 2015 and 2025 were identified across PubMed, Scopus, and Google Scholar. Eligible studies included dietary, nutritional, or supplement-based interventions reporting muscle-related outcomes in obesity- or diabetes-associated conditions. Results: Ninety-one eligible studies were categorized into plant-derived, animal/marine-based, microorganism/fermented, synthetic/pharmaceutical, and environmental interventions. Across diverse models, bioactive compounds such as D-pinitol, umbelliferone, resveratrol, GABA, ginseng, whey peptides, probiotics, and omega-3 fatty acids consistently improved muscle mass, strength, and mitochondrial function via AMPK–SIRT1–PGC-1α and Akt–mTOR signaling. These mechanisms promoted mitochondrial biogenesis, suppressed proteolysis (MuRF1, Atrogin-1), and enhanced insulin sensitivity, antioxidant capacity, and gut–muscle communication. Conclusions: PHD-aligned foods combining plant proteins, polyphenols, and fermented products strengthen nutrient sensing, mitochondrial efficiency, and cellular resilience, representing a sustainable nutritional framework for preventing and managing SO and DS.

## 1. Introduction

One of the most profound demographic and societal shifts in the 21st century is the aging of the global population. Advances in medical science, public health measures such as improved sanitation, widespread vaccination, the use of antibiotics, and better maternal and child healthcare, along with socioeconomic progress, have together driven an unparalleled rise in human life expectancy [1]. It is predicted that by 2030, one in six people globally will be 60 or older, with this group growing from 1 billion in 2020 to 1.4 billion, and the population aged 80 and above is projected to triple by 2050 [2].

As people age, they experience progressive declines in skeletal muscle mass, metabolism, and functional capacity, which are linked to reduced independence and increased mortality [3]. This gradual loss of muscle mass and function is called sarcopenia, a progressive skeletal muscle disorder marked by a faster decline in muscle quantity and strength, which the Asian Working Group for Sarcopenia (AWGS) 2019 consensus defines as “age-related loss of muscle mass, plus low muscle strength, and/or low physical performance.” [4,5].

The term sarcopenia was first introduced by Irwin Rosenberg in 1988, derived from the Greek words *sarx*, meaning “flesh,” and *penia*, meaning “loss” [6]. It predominantly affects older adults and is associated with increased risks of falls, functional decline, frailty, and mortality. It typically begins around the age of 50, with muscle fiber numbers steadily declining so that by age 80, nearly half are lost [7]. Many factors contribute to the development of sarcopenia, with the primary instigators including malnutrition, physical inactivity, cellular senescence, and various chronic diseases [8].

In addition to muscle deterioration, physiological aging is often accompanied by excess fat accumulation and insulin resistance, which significantly increase the risk of chronic conditions such as obesity and type 2 diabetes [9]. When insulin resistance happens, it reduces glucose uptake, while excess fat releases pro-inflammatory cytokines that accelerate muscle loss, which in turn further decreases insulin-responsive tissue and worsens insulin resistance, creating a vicious cycle that contributes to sarcopenic obesity (SO) and diabetic sarcopenia (DS) [10].

SO is characterized by excessive adiposity coexisting with diminished skeletal muscle quantity and quality [11]. A key pathological feature of this condition is myosteatosis, which involves the infiltration of adipose tissue into skeletal muscle. Myosteatosis promotes lipotoxicity, oxidative stress, and mitochondrial dysfunction, which disrupt insulin signaling and muscle metabolism, ultimately impairing muscle regeneration and quality in SO [12]. Excess macronutrient storage in adipose tissue also stimulates pro-inflammatory cytokines, such as tumor necrosis factor-α and interleukin-6, while decreasing adiponectin, further aggravating systemic inflammation and oxidative stress [13].

Meanwhile, in DS, chronic hyperglycemia and insulin resistance accelerate skeletal muscle atrophy through multiple intersecting pathways [14]. Insulin resistance, a key pathological feature of type 2 diabetes mellitus (T2DM), not only impairs glucose utilization but also compromises the anabolic capacity of skeletal muscle [15]. Normally, insulin facilitates amino acid uptake into muscle cells, promoting protein synthesis and preserving muscle integrity [16]. This occurs via canonical insulin signaling, which activates the Akt/mTOR pathway to enhance protein synthesis and inhibits protein degradation through the Akt/FOXO pathway [17]. In insulin resistance, these anabolic pathways are impaired, while catabolic systems such as the ubiquitin–proteasome and calpain-dependent pathways are overactivated, leading to accelerated muscle protein breakdown [18].

While aging establishes a physiological baseline for these degenerative processes, sarcopenia can be further accelerated by disease-driven metabolic dysfunction. In particular, SO and DS represent compounded states where age-related muscle loss is intensified by pathological mechanisms associated with obesity and diabetes, respectively [14,19]. These conditions arise from shared underlying mechanisms, including chronic inflammation, insulin resistance, physical inactivity, and hormonal changes, leading to a worsening cycle that negatively impacts health, function, and increases risks for other chronic diseases [10].

At the molecular level, these conditions are associated with anabolic resistance, mitochondrial dysfunction, oxidative stress, chronic low-grade inflammation, and impaired proteostasis [20,21]. Proteostasis, or protein homeostasis, encompasses the regulated processes of protein synthesis, folding, and degradation that preserve skeletal muscle integrity. Disruption of this balance leads to the buildup of misfolded proteins, contributing to muscle atrophy and metabolic dysfunction [22]. Molecular chaperones such as HSP60, HSP70, and HSP90 play essential roles in maintaining cellular homeostasis by ensuring proper protein folding, preventing aggregation, and promoting the degradation or repair of damaged proteins, thereby protecting tissues from inflammation and injury [23]. Beyond chaperone-mediated protein folding, cells maintain proteostasis through multiple degradation pathways, including the ubiquitin–proteasome system, the autophagy–lysosome pathway, and the unfolded protein response, which collectively eliminate misfolded or damaged proteins and preserve cellular quality control [22,24]. Dysregulation of these systems, together with impaired activation of anabolic signaling pathways such as IGF-1/Akt/mTOR and excessive activity of catabolic regulators including FoxO3a, MuRF1, and Atrogin-1, further accelerates muscle degradation and functional decline [15,20,24].

Currently, there are no pharmacological treatments available to prevent sarcopenia, slow its progression, or lessen its health risks. The best way to combat sarcopenia is by adopting lifestyle changes, especially maintaining a nutrient-rich diet and participating in consistent physical activity [25]. A higher intake of vegetables and fruits is associated with a reduced risk of sarcopenia, and consuming natural, nutrient-rich foods containing high-quality protein, fruits, and vegetables may help prevent muscle wasting and age-related muscle loss [26]. In addition, nutritional supplementation, including essential amino acids, vitamins, polyphenols, and functional ingredients derived from probiotics, prebiotics, and postbiotics, has shown potential to enhance mitochondrial function, reduce inflammation, and improve protein turnover, thereby promoting muscle preservation and metabolic resilience [27,28,29,30].

Poor dietary habits are key drivers of chronic diseases, including obesity, heart disease, diabetes, and early death [31]. As a result, dietary approaches that address both human health and environmental sustainability have received increasing attention in recent years. One of the most widely discussed frameworks is the Planetary Health Diet (PHD), proposed by the EAT–Lancet Commission in 2019 [32]. The planetary health diet recommends that about half of the plate be filled with fruits and vegetables, while the remaining half should mainly include whole grains, plant-based proteins, and unsaturated oils, with the option of adding small portions of animal-source foods [33]. This dietary model is intended to improve population health by lowering the risk of chronic diseases while at the same time advancing environmental sustainability [31]. Dietary strategies to counteract sarcopenia should not only ensure adequate calories and nutrients but also emphasize overall nutritional patterns. Within the Planetary Health Diet framework, such approaches could assist in characterizing metabolic responses relevant to SO and DS, including protein synthesis, inflammation, and oxidative stress [25]. Accordingly, this review aims to systematically examine the correlations of SO and DS with dietary patterns, particularly within the planetary health diet framework.

## 2. Materials and Methods

### 2.1. Protocol and Registration

This systematic review was conducted in accordance with the Preferred Reporting Items for Systematic Reviews and Meta-Analyses (PRISMA 2020) guidelines to ensure methodological transparency and reproducibility. The protocol for this review was prospectively registered in the International Prospective Register of Systematic Reviews (PROSPERO; registration ID: CRD420251127098). The primary objective of this review was to evaluate and compare the dietary recommendations and intervention outcomes associated with the Planetary Health Diet (PHD) in the prevention and management of sarcopenic obesity (SO) and diabetic sarcopenia (DS).

### 2.2. Eligibility Criteria

Studies were considered eligible if they investigated human or animal models relevant to SO or DS and were published between January 2015 and September 2025. Only studies written in English or with accessible English translations were included. Eligible study designs comprised randomized controlled trials, controlled clinical trials, and cohort studies that reported at least one muscle-related outcome, such as skeletal muscle mass, muscle strength, or physical performance.

Publications were excluded if they were reviews, editorials, commentaries, conference abstracts, or studies that did not provide relevant muscle-related outcomes. The selection of studies was conducted by one reviewer and independently verified by a second reviewer to ensure accuracy and consistency. Any discrepancies in the inclusion decision were resolved through discussion, and no automation tools were used in the selection process.

### 2.3. Information Sources and Search Strategy

A comprehensive literature search was performed using three major electronic databases: PubMed, Scopus, and Google Scholar. The search covered all available records up to 30 September 2025. The search strategy combined terms such as “Planetary Health Diet,” “EAT–Lancet,” “plant-based diet,” “vegan,” “vegetarian,” “sarcopenia,” “sarcopenic obesity,” “diabetic sarcopenia,” “muscle strength,” “muscle mass,” “hand grip,” “gait speed,” and “SPPB.” Filters were applied to restrict results to the English language, and publications from 2015 onward. In addition, manual searches of the reference lists of relevant papers and reviews were conducted to identify any additional eligible studies that were not captured through database searching.

### 2.4. Data Extraction and Items

Data extraction was performed using a standardized form developed by the research team to ensure consistency across studies. Extracted data included details on skeletal muscle mass, muscle strength, and physical performance, along with relevant metabolic outcomes such as insulin sensitivity, lipid profiles, and inflammatory markers. Information regarding study design, sample size, participant characteristics (age, sex, and health status), intervention type, intervention duration, and dosage was also collected. When available, proposed mechanistic pathways and funding sources were recorded to contextualize study findings. Any missing or ambiguous data were interpreted cautiously based on the information provided in the original publication, and no assumptions were made beyond what the authors explicitly reported.

### 2.5. Assessment of Study Quality

Because the purpose of this review was qualitative and exploratory, no formal risk-of-bias scoring tools were applied. Instead, the methodological quality of each study was assessed narratively, focusing on factors such as the appropriateness of the study design, adequacy of the sample size, transparency in outcome reporting, and internal consistency of findings. Studies were also evaluated for clarity in describing intervention protocols and the reproducibility of their reported results.

### 2.6. Data Synthesis and Analysis

All eligible studies were grouped according to the type of dietary intervention investigated. Categories included plant-derived interventions, animal or marine-based interventions, microbial or fermented products, synthetic or pharmaceutical compounds, and environmental or combined dietary approaches. Findings were summarized in comparative tables to highlight differences and similarities across intervention types. Narrative synthesis was applied to describe the direction, magnitude, and consistency of observed effects rather than to perform statistical pooling. Figures and summary tables were used to visualize key mechanistic pathways and outcome trends. Because of substantial heterogeneity in study design, population characteristics, and outcome measures, no meta-analysis, subgroup analysis, or sensitivity analysis was conducted. Confidence in the overall body of evidence was assessed qualitatively, considering the consistency, transparency, and quality of the included studies. No formal risk-of-bias or certainty-of-evidence frameworks (such as ROBIS or GRADE) were applied due to the considerable heterogeneity across study designs, models, and intervention characteristics. Instead, reliability and confidence in the evidence were evaluated descriptively, based on methodological clarity, reporting completeness, and coherence of findings across studies.

### 2.7. Ethical Considerations

This study synthesized information exclusively from previously published research and therefore did not involve any human or animal experimentation. Ethical approval and informed consent were not required.

## 3. Results

Following the PRISMA workflow, our systematic search retrieved 718 records from PubMed, Scopus, and Google Scholar. After removal of 41 duplicates, 677 records were screened by title and abstract. Studies were excluded for the following reasons: (1) not original research (e.g., reviews, commentaries, conference abstracts); (2) not related to sarcopenic obesity or diabetic sarcopenia; (3) no dietary or nutritional intervention; (4) in vitro or animal studies without mechanistic relevance; or (5) insufficient or non-extractable outcome data. The full texts of potentially eligible articles were then assessed, and 91 studies met the inclusion criteria for qualitative synthesis (Figure 1).

A total of 91 studies were included in this systematic review, as illustrated in Figure 1. These studies were categorized into four major intervention sources based on their bioactive origins. Plant-derived compounds (Table 1), such as D-pinitol, umbelliferone, resveratrol, γ-aminobutyric acid (GABA), and ginseng, consistently enhanced mitochondrial biogenesis, insulin sensitivity, and protein synthesis through activation of the AMPK–SIRT1–PGC-1α and Akt–mTOR signaling pathways. Animal- and marine-based interventions (Table 2), including whole-egg feeding, whey peptides or Sarcomeal^®^ supplementation, krill oil, and melatonin combined with exercise, promoted lean mass and muscle strength via mTOR activation and improved amino acid transport. Microorganism- and fermented-product interventions (Table 3), such as probiotics, postbiotics, and colostrum-derived exosomes, improved muscle quality by modulating the gut–muscle axis and restoring AMPK–SIRT1–PGC-1α signaling. Finally, environmental and stress-modulating strategies (Table 4), including exercise, thermal manipulation, and omega-3 fatty acid intake, supported proteostasis and antioxidant defense through heat shock protein (HSP)-mediated pathways.

## 4. Discussion

In this systematic review, we identified consistent evidence that bioactive nutrients aligned with the Planetary Health Diet enhance muscle mass, strength, and metabolic resilience in sarcopenic obesity and diabetic sarcopenia through several convergent biological mechanisms. Specifically, most interventions activated the AMPK–SIRT1–PGC-1α axis, enhanced Akt–mTOR-mediated protein synthesis, suppressed FOXO-driven proteolysis, improved mitochondrial quality control, reduced inflammation and oxidative stress, and restored gut–muscle communication. These shared pathways highlight that nutrient-dense plant-forward dietary components may simultaneously target metabolic dysfunction and muscle degeneration, suggesting a coherent mechanistic basis for dietary strategies in managing SO and DS.

### 4.1. Pathophysiology and Metabolic Dysregulation in Sarcopenia Risk Factors

Sarcopenia can be broadly categorized into primary and secondary types. Primary sarcopenia develops with advancing age as part of the natural aging process, leading to a gradual and irreversible decline in muscle mass and strength. In contrast, secondary sarcopenia arises from underlying conditions such as chronic diseases, malignancies, malnutrition, or metabolic imbalances [65]. At the cellular level, sarcopenia is characterized by disrupted protein homeostasis, changes in epigenetic regulation, loss of satellite cells, abnormal fat accumulation within muscle, a decline in myofiber number, genomic instability, programmed muscle cell death, and telomere shortening [66]. These hallmarks reflect the cumulative damage and functional decline in skeletal muscles, which is further influenced by multifactorial interactions among intrinsic aging, chronic disease, and lifestyle factors, as illustrated in Figure 2. Figure 2 illustrates the multifactorial risk factors that contribute to the development of sarcopenia, including aging, chronic diseases such as COPD, cancer, HIV, diabetes, and kidney disease, as well as malnutrition and genetic predispositions. Lifestyle-related contributors—including physical inactivity, obesity, insulin resistance, and rheumatoid arthritis—further accelerate muscle loss and functional decline. Together, these biological, nutritional, and behavioral influences interact to increase vulnerability to sarcopenia.

Clinically, the coexistence of metabolic dysregulation and muscle degradation poses significant health challenges [67]. Individuals with SO or DS face markedly higher risks of frailty, falls, insulin resistance, disability, and mortality compared to those with sarcopenia or obesity alone [68]. Impaired glucose and lipid metabolism worsens muscle loss, while reduced muscle mass further impairs glycemic control, creating a vicious metabolic–musculoskeletal feedback loop that complicates disease management and rehabilitation [69].

A growing body of evidence, including the present synthesis, highlights dysregulated nutrient sensing as a central mechanism linking metabolic disorders to muscle wasting in sarcopenic obesity (SO) and diabetic sarcopenia (DS) [24,70]. Under physiological conditions, nutrient sensors such as AMP-activated protein kinase (AMPK), sirtuin-1 (SIRT1), and mammalian target of rapamycin (mTOR) orchestrate the balance between anabolism and catabolism in response to energy and substrate availability [53,71]. AMPK and SIRT1 act as energy sensors that activate mitochondrial biogenesis and fatty acid oxidation, whereas mTOR integrates amino acid and insulin signals to promote protein synthesis [28,36]. Together, these pathways ensure efficient energy utilization, proteostasis, and muscle regeneration.

In metabolic disease states characterized by chronic nutrient excess, lipotoxicity, and insulin resistance, this regulatory network becomes disrupted [72]. Persistent nutrient oversupply inhibits AMPK and SIRT1 activity, while causing abnormal mTOR activation. This leads to reduced autophagy, mitochondrial problems, and anabolic resistance, which is the decreased capacity of skeletal muscle to produce proteins despite sufficient nutrient intake [10,30].

As illustrated in Figure 3, muscle mass is regulated by the dynamic balance between muscle protein synthesis (MPS) and muscle protein breakdown (MPB). Figure 3 illustrates the regulation of muscle protein turnover, showing that muscle protein synthesis (MPS) is stimulated by anabolic factors such as leucine, exercise, and insulin through activation of the mTOR pathway, while muscle protein breakdown (MPB) is promoted by catabolic stimuli—including aging, disease, inflammation, and AMPK activation—primarily through the upregulation of the E3 ubiquitin ligases MuRF-1 and Atrogin-1. In a controlled trial by McKendry et al. (2024) [51], dietary protein intake, meal distribution, and physical activity were standardized to isolate the effects of protein source and dose on MPS. Their findings demonstrated that protein quantity, quality, and leucine content are determinants of anabolic signaling through the mTORC1–rpS6 pathway. However, inadequate or poorly distributed protein intake blunts MPS and tilts the balance toward proteolysis, accompanied by reduced IGF-1/Akt/mTOR signaling, elevated proteolytic activity (MuRF-1 and Atrogin-1), and mitochondrial deterioration as observed in diabetic and obese muscle [28,43,71,73].

### 4.2. Findings and Evidence Trends from Reviewed Studies

Across the reviewed literature, several consistent physiological benefits were reported. Most interventions improved muscle quantity and quality, as reflected by increases in lean body mass, enlargement of muscle fiber cross-sectional area, enhanced grip strength, and improvements in muscle performance parameters such as endurance or locomotor activity [24,25,26,27,28,29,30,31,32,33,34,35,36,37,38,39,40,41,42,43,44,45,46,47,48,49,50]. Although a wide range of experimental models were used, the observed benefits consistently converged on several closely linked biological pathways governing energy metabolism, protein homeostasis, and redox regulation. Figure 4 illustrates the key mechanistic pathways through which nutritional and lifestyle interventions may counteract SO and DS, including suppression of FOXO-mediated proteolysis, enhancement of anti-inflammatory and antioxidant defenses, improvement of mitophagy and mitochondrial quality control, support of heat shock protein–mediated proteostasis, restoration of the gut–muscle axis, activation of the AMPK–SIRT1–PGC-1α signaling cascade, and stimulation of Akt–mTOR anabolic signaling. While these results highlight mechanistic pathways, most evidence is from animal and cell studies, making clinical relevance difficult to determine. The review aimed to be comprehensive, but variations in database indexing and terminology may have limited some study retrieval. These findings suggest that Planetary Health Diet-aligned eating patterns could support muscle and metabolic health, highlighting the value of sustainable, nutrient-rich foods in managing SO and diabetes. Future research should confirm these effects in human trials and assess long-term adherence and real-world applicability.

#### 4.2.1. AMPK–SIRT1–PGC-1α Axis Activation

The most frequently discussed mechanism is the AMPK/SIRT1/PGC-1α signaling pathway, a central regulator of mitochondrial biogenesis and the cellular response to oxidative stress [68]. When intracellular energy levels decline, indicated by an increased AMP/ATP ratio, AMPK becomes activated and triggers downstream cascades to restore energy balance [65]. In skeletal muscle, AMPK works synergistically with another metabolic sensor, SIRT1, which deacetylates downstream targets such as PGC-1α to enhance mitochondrial activity and overall metabolic function [21]. PGC-1α, in turn, promotes mitochondrial replication and biogenesis, driving the remodeling of muscle fibers toward a more oxidative and less glycolytic phenotype, thereby improving metabolic efficiency and endurance capacity [37].

#### 4.2.2. Akt–mTOR Signaling Enhancement

The mTOR serves as a central regulator of cellular processes, controlling protein synthesis, growth, proliferation, autophagy, lysosomal activity, and overall metabolism. Dysregulation of mTOR signaling has been linked to a range of pathological conditions, including cardiovascular diseases, cancer, and metabolic disorders [69]. Two anabolic signaling routes were commonly found. Insulin-like growth factor-1 (IGF-1) signaling promotes muscle growth by activating the Akt/mTOR pathway, which enhances protein synthesis and contributes to the maintenance of muscle mass [74]. Activation of this pathway not only stimulates anabolic processes but also suppresses protein degradation by inhibiting the ubiquitin–proteasome system (UPS) and the autophagy–lysosome pathway (ALP) [21]. In parallel, the PI3K/Akt/mTOR (PAM) pathway functions as a nutrient-sensing network that integrates amino acid availability, particularly leucine, to activate mTORC1 and downstream effectors such as p70S6K and 4E-BP1, driving protein translation and muscle growth [74,75].

#### 4.2.3. FOXO Inhibition and Proteolysis Suppression

The Forkhead box O (FoxO) family of transcription factors, which is indirectly suppressed by insulin signaling, plays a key role in regulating protein catabolism by activating proteolytic systems [19]. When insulin or IGF-1 signaling is impaired, Akt activity declines, leading to FOXO nuclear translocation and increased expression of the E3 ubiquitin ligases Muscle RING-finger 1 (MuRF1) and Atrogin-1 (MAFbx), which target structural proteins such as myosin and actin for degradation [19,65,74]. By inhibiting the activation of Forkhead box O (FOXO) proteins or promoting their phosphorylation-dependent sequestration in the cytoplasm, their nuclear translocation is prevented. As a result, FOXO transcription factors are unable to induce the expression of proteolysis-related genes, including the E3 ubiquitin ligases muscle RING-finger protein-1 (MuRF-1) and muscle atrophy F-box (MAFbx), thereby suppressing muscle protein degradation [24].

#### 4.2.4. Mitophagy and Mitochondrial Quality Control

Mitochondrial quality control (MQC) involves an integrated network of mitochondrial biogenesis, dynamics (fusion and fission), and selective autophagy (mitophagy) that collectively maintain mitochondrial integrity and function [76]. Under normal physiological conditions, MQC eliminates damaged mitochondria, limits excessive reactive oxygen species production, and regulates apoptosis [76,77]. When mitochondria become damaged and lose membrane potential, PINK1 accumulates on the outer mitochondrial membrane, where it phosphorylates ubiquitin and Parkin, activating Parkin’s E3 ubiquitin ligase activity and recruiting it to the dysfunctional organelle. Activated Parkin then ubiquitinates outer membrane proteins, marking the mitochondrion for selective autophagy and, in some cases, directing ubiquitinated substrates toward proteasomal degradation [78]. In the context of SO and DS, chronic metabolic stress severely disrupts mitochondrial quality control [79,80]. Reduced expression of fusion proteins (MFN1/2, OPA1) and increased fission (DRP1, FIS1) promote excessive fragmentation, while impaired PINK1 stabilization and Parkin recruitment hinder mitophagy [28,36]. As a result, damaged mitochondria accumulate, generating excess ROS, releasing pro-apoptotic factors, and exacerbating insulin resistance and muscle atrophy [81].

#### 4.2.5. Anti-Inflammatory and Antioxidant Modulation

SO and DS share a chronic pro-inflammatory and oxidative environment driven by adipose-derived cytokines and hyperglycemia-induced ROS [10]. During this, adipose tissue amplifies inflammation by releasing pro-inflammatory cytokines such as TNF-α, IL-6, and IL-1β, which further promote ROS generation [14,22,82]. Both obesity and hyperglycemia stimulate the release of these mediators, activating macrophages and other immune cells, as well as apoptosis-related pathways such as Fas/FasL signaling, thereby exacerbating tissue damage and muscle degradation [10]. Bioactive compounds such as *resveratrol*, *curcumin*, *anthocyanins*, and *propolis* disrupt this loop by scavenging free radicals, activating Nrf2-dependent antioxidant enzymes (SOD, CAT, GPx), and inhibiting NF-κB [29,44,45]. The result is decreased cytokine burden, improved insulin sensitivity, and restoration of redox equilibrium.

#### 4.2.6. Gut–Muscle Axis Restoration

The gut microbiome plays a vital role in keeping skeletal muscle healthy through a network of metabolic, immune, and hormonal signals, often referred to as the gut–muscle axis [83]. When the normal balance of gut bacteria is disturbed, known as microbial dysbiosis, the body produces fewer beneficial substances such as short-chain fatty acids (SCFAs) and branched-chain amino acids (BCAAs) [84]. These substances typically help muscles generate energy, support mitochondrial function, and reduce inflammation [57]. When their levels decrease, energy production in muscle cells becomes less efficient and inflammation increases throughout the body. Over time, this imbalance can weaken muscles and contribute to the development of sarcopenia. Dysregulation of the gut microbiota can also contribute to the development or progression of sarcopenia and obesity by altering the expression of myostatin and atrogin-1, as well as disrupting the signaling between the enteric nervous system and the brain. These alterations, in turn, negatively affect muscle mass and appetite [48].

Commensal taxa such as Faecalibacterium, Roseburia, and members of the Lachnospiraceae family contribute to the maintenance of intestinal barrier integrity, reduction in circulating lipopolysaccharide (LPS), and activation of the AMPK/PGC-1α signaling axis, which facilitates mitochondrial biogenesis and supports muscle protein synthesis and turnover [61]. In contrast, the depletion of these beneficial microbes is associated with the upregulation of catabolic transcription factors, including NF-κB and FOXO3a, leading to muscle protein degradation and atrophy [57]. Intervention strategies that target the gut microbiota, such as the administration of probiotics, dietary polyphenols, or microbiota-derived exosomes, have demonstrated efficacy in restoring microbial composition, enhancing anti-inflammatory metabolite production, and re-sensitizing peripheral tissues to insulin [48,49]. Therefore, restoration of the gut–muscle axis extends beyond mere hypertrophy; it represents a systemic recalibration of host metabolic and inflammatory states, highlighting the microbiome’s pivotal role in both preventing and reversing sarcopenia.

#### 4.2.7. Heat Shock Protein and Proteostasis Support

Heat shock proteins (HSPs) constitute a highly conserved family of molecular chaperones that play a central role in maintaining cellular proteostasis [85]. They assist in the correct folding of nascent polypeptides, prevent protein aggregation under stress conditions, and facilitate the refolding or degradation of damaged proteins through the ubiquitin–proteasome and autophagy–lysosome systems [86]. In metabolic disorders such as SO and DS, chronic low-grade inflammation and oxidative stress markedly suppress HSP expression and function [87]. This downregulation disrupts the proteostatic network, leading to the accumulation of misfolded proteins, endoplasmic reticulum (ER) stress, and activation of apoptotic pathways, thereby impairing myocyte survival and regeneration [10]. Reduced HSP70 and HSP90 levels weaken mitochondrial stability and autophagic clearance, promoting metabolic and structural decline. Conversely, interventions such as moderate exercise, heat therapy, and bioactive compounds, such as curcumin or resveratrol, can reactivate HSP expression via HSF1-AMPK-SIRT1 pathways, restoring proteostasis and enhancing muscle resilience [88]. Strengthening HSP-mediated defenses thus provides a holistic strategy that integrates redox balance, mitochondrial quality, and cellular repair mechanisms. Collectively, supporting HSP-driven cytoprotection represents a promising therapeutic direction for preserving skeletal muscle structure and metabolic function [89]. By bridging redox regulation, proteostasis, mitochondrial quality control, and metabolic signaling, the HSP–HSF1–AMPK–SIRT1 axis emerges as a pivotal adaptive target to mitigate the progressive decline associated with SO and DS.

### 4.3. Whole-Diet and Nutritional Approaches in Managing SO and DS

The compiled interventions suggest that many effective compounds for SO and DS may be derived from common foods or traditional dietary ingredients rather than synthetic agents. This indicates that the potential therapeutic value of these bioactives could be achieved through strategic whole diet composition, emphasizing nutrient-dense, minimally processed foods that may deliver synergistic molecules acting on mitochondrial, inflammatory, and proteostatic pathways.

#### 4.3.1. Legumes and Soy-Derived Foods

Legumes, including soybeans, peas, and Vigna species, are considered key sources of bioactive compounds such as D-pinitol and isoflavones, as well as high-quality plant proteins. These nutrients may help restore mitochondrial biogenesis and improve insulin sensitivity through MFG-E8 inhibition and mitophagy activation, while potentially modulating the gut microbiota toward increased Lachnospiraceae and other short-chain fatty acid (SCFA) producers [34,35]. Among legume proteins, pea protein was found to provide a high-quality plant source that supports muscle protein synthesis as effectively as whey, helping maintain muscle mass and metabolic health in older adults [51]. Regular consumption of soy-derived foods such as tofu, tempeh, soy milk, and edamame can support muscle anabolism and metabolic balance, providing sustainable protein sources that align with PHD principles. Similarly, L-arabinose, a plant-derived sugar abundant in legume and cereal cell walls, improved glucose metabolism and reduced inflammation, suggesting that functional carbohydrates can reinforce the anti-diabetic and anti-sarcopenic potential of PHD-aligned diets [38].

#### 4.3.2. Vegetables, Herbs, and Polyphenol-Rich Plants

Coumarins and polyphenols present in common vegetables and herbs may also influence pathways related to muscle quality. Umbelliferone, present in carrots, parsley, celery, and fennel, increased type I muscle-fiber proportion and reduced lipid accumulation through AMPK–SIRT1–PGC-1α and mitochondrial-fusion proteins (MFN1/OPA1) [36]. Extracts from *Lespedeza bicolor*, *Codonopsis lanceolata*, and *Setaria viridis also enhanced muscle mass and antioxidant capacity by activating AMPK/SIRT1/PGC-1α and Akt/mTOR signaling* [37,42,43]. These species, traditionally used in East Asia as herbal teas, condiments, or functional extracts rather than staple foods, illustrate that bioactive plant compounds studied in experimental models may reflect mechanisms that could also occur at lower levels in ordinary vegetables and herbs.

#### 4.3.3. Fruits and Fermented Plant Products

Polyphenol-rich fruits and fermented plant products may improve mitochondrial and proteostatic functions. Resveratrol, abundant in grapes and berries, has been shown to reduce intramyocellular lipid deposition and enhance PKA/LKB1/AMPK–PGC-1α signaling while potentially lowering Atrogin-1 and MuRF-1 expression [28,45]. Aged black garlic has been reported to increase myogenic markers (MyoD, Myogenin, MRF4) and stimulate Akt/mTOR/p70S6K signaling, suggesting promotion of protein synthesis and mitochondrial biogenesis [46]. These results indicate that naturally occurring antioxidant and redox-active compounds in fruits and fermented allium products may influence muscle maintenance.

#### 4.3.4. Animal- and Marine-Based Protein Sources

Although plant-forward diets are central, moderate inclusion of whole eggs, whey peptides, and marine oils may provide complementary anabolic support. Studies using whole eggs, fermented soy, and dairy-derived peptides have suggested improvements in muscle mass and strength, potentially via activation of Akt/mTOR signaling and suppression of proteolytic enzymes [30,52,53]. These findings support the inclusion of modest amounts of high-quality animal proteins within a plant-forward framework to maintain muscle anabolic responsiveness in aging and diabetes.

#### 4.3.5. Fermented and Microbiota-Active Foods

Fermented foods and dietary fibers play a key role in maintaining a healthy microbiome, which interacts with skeletal muscle through the gut–muscle axis [90]. Experimental and clinical data suggest that probiotic and postbiotic interventions involving Lactobacillus and Bifidobacterium species can improve muscle fiber quality, insulin sensitivity, and mitochondrial activity, potentially via AMPK–SIRT1–PGC-1α and Akt–mTOR pathways [57,58,59,60,61]. In dietary terms, these effects correspond to traditional fermented foods, such as yogurt, kefir, kimchi, miso, tempeh, or natto, which can foster SCFA-producing taxa such as Faecalibacterium and Roseburia [59,91]. Regular consumption of these foods could help maintain intestinal integrity and metabolic signaling relevant to muscle health.

#### 4.3.6. Practical Applications and Limitation

Although plant-forward dietary patterns such as the Mediterranean, vegetarian, vegan, and various regionally specific diets differ in their profiles of bioactive nutritional components, there is currently insufficient evidence directly comparing these dietary models in relation to SO or DS. Research comparing the Mediterranean, vegetarian, vegan, or Eastern dietary patterns within this context remains very limited, and studies evaluating the Planetary Health Diet as a structured model for these conditions are likewise scarce. Therefore, this review focuses on identifying foods, food groups, and bioactive components that influence metabolic and muscle-related pathways relevant to SO and DS, rather than comparing entire dietary patterns. Nonetheless, these findings may help inform practical lifestyle strategies by highlighting nutritional elements that can be integrated into different eating patterns, including those aligned with the PHD, to support muscle health in at-risk populations.

In practical terms, these findings suggest that incorporating nutrient-dense plant foods—such as legumes, nuts, whole grains, mushrooms, leafy vegetables, and polyphenol-rich fruits—may support muscle metabolism by enhancing mitochondrial function, reducing inflammation, and promoting protein synthesis. Individuals with SO or DS may particularly benefit from balanced meal patterns that emphasize adequate protein distribution, leucine-rich plant proteins, and antioxidant- or fiber-rich foods that modulate gut–muscle signaling. From a clinical perspective, dietitians and healthcare practitioners can use these mechanistic insights to design culturally adaptable meal plans rooted in PHD principles, improving patient adherence while addressing both metabolic and muscle-related impairments. These actionable dietary strategies may be integrated into existing lifestyle and chronic disease management programs to support functional health and reduce progression of sarcopenia-related complications.

In addition, the evidence base is constrained by substantial heterogeneity across studies, including differences in the dose, duration, and delivery of nutritional interventions, which limits the comparability of findings. Furthermore, only a small proportion of studies were conducted in humans, with most data derived from cell and animal models. This imbalance restricts the direct translation of mechanistic observations to clinical practice and underscores the need for well-designed human trials.

## 5. Conclusions

This review highlights that nutrient-dense foods commonly featured in the Planetary Health Diet may support muscle and metabolic health in individuals with sarcopenic obesity or diabetic sarcopenia. Although the available evidence suggests consistent benefits across multiple nutritional components, most studies focus on isolated foods or compounds rather than whole dietary patterns. Direct evaluations of the PHD and other plant-forward dietary models in SO or DS remain limited. Future human studies are needed to determine whether specific dietary patterns offer distinct advantages and to establish practical, culturally adaptable dietary strategies for clinical application.

## Figures and Tables

**Figure 1 nutrients-17-03656-f001:**
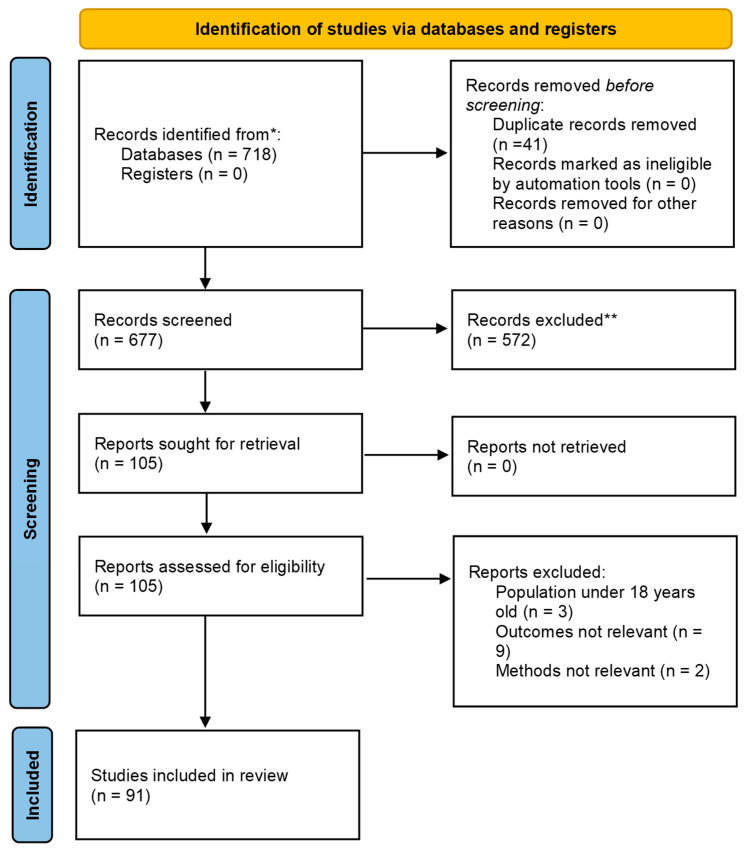
PRISMA 2020 flow diagram of study selection process for the systematic review. * Records identified from electronic databases after removing duplicates. ** Records excluded after title/abstract screening based on irrelevance to outcomes, population, or study type.

**Figure 2 nutrients-17-03656-f002:**
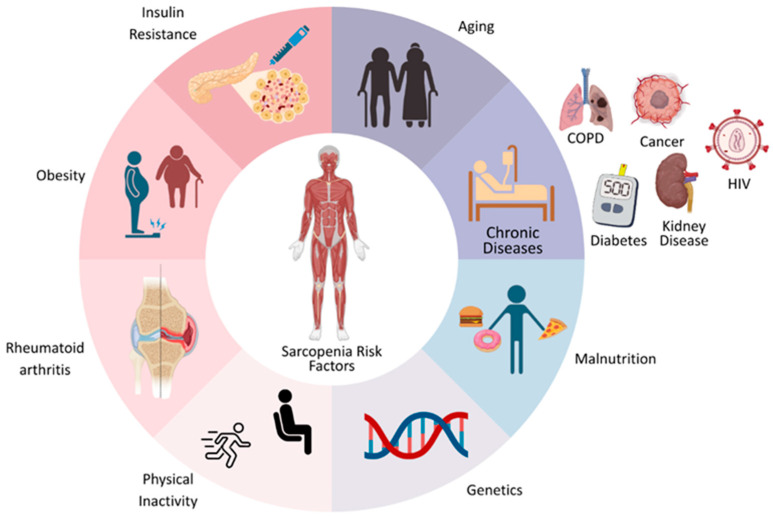
Multifactorial risk factors contributing to the development of sarcopenia.

**Figure 3 nutrients-17-03656-f003:**
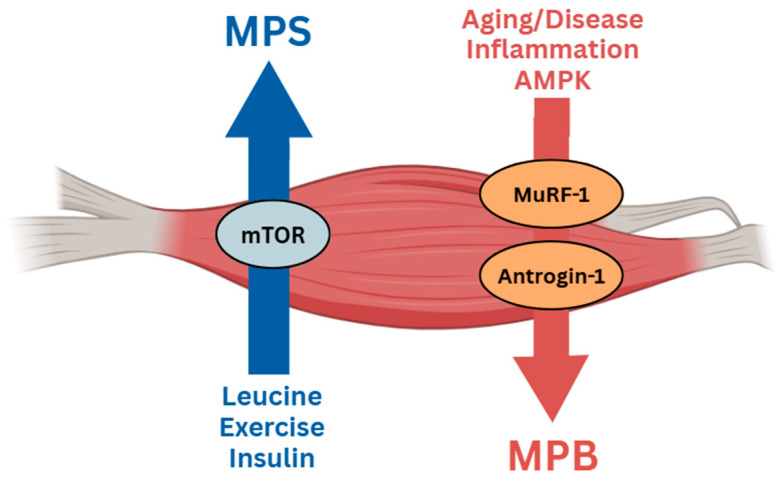
Regulation of Muscle Protein Synthesis (MPS) and Muscle Protein Breakdown (MPB).

**Figure 4 nutrients-17-03656-f004:**
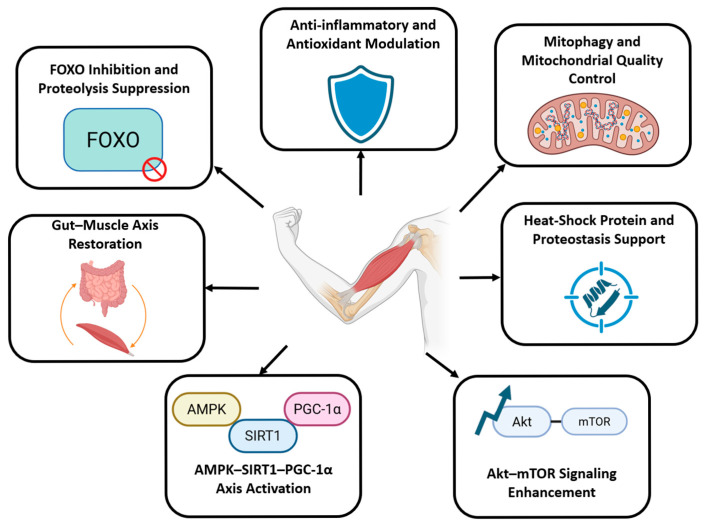
Mechanistic pathways and evidence trends were identified across reviewed interventions targeting sarcopenic obesity (SO) and diabetic sarcopenia (DS). The red prohibition symbol in the lower right corner represents the inhibition or suppression of FOXO activity.

**Table 1 nutrients-17-03656-t001:** Overview of Plant-Derived Interventions for Sarcopenic Obesity and Diabetic Sarcopenia.

Intervention	Source	Study Type	Population/Model	Dose and Duration	Muscle Outcomes	Metabolic Outcomes	Mechanism	Ref.
D-pinitol targeting MFG-E8	Soybean seeds	Animal and Cell	Mice: Older diabetic mice with muscle loss. Groups: (1) diabetic mice untreated, (2) diabetic mice given D-pinitol Cells: C2C12 muscle cells stressed with advanced glycation end-products (AGEs) or D-galactose. Groups: (1) stressed cells untreated, (2) stressed cells treated with D-pinitol.	Mice: Given D-pinitol 150 mg/kg/day by mouth for 6 weeks. Cells: Treated with D-pinitol 160 µM for 48 h.	Muscle strength: ↑ grip strength, ↑ endurance Muscle mass: ↑ lean mass, ↑ muscle weight, ↑ fiber size Muscle function: improved mitochondrial structure; in cells: ↑ survival, ↓ damage, restored mitophagy	Diabetes: ↓ blood glucose	Mitochondrial QC/Mitophagy: ↑ PINK1, ↑ Parkin, ↑ LC3B, ↓ P62 (restored mitophagy). Proteostasis/HSP: MFG-E8 suppression → release of HSPA1L–Parkin axis.	[34]
D-pinitol	Vigna sinensis, soybean, and pine plants.	Animal	STZ-induced SAMP8 diabetic aging mice (*n* = 10–12/group)	150 mg/kg/day orally for 8 weeks	Muscle strength: ↑ grip strength Muscle mass: ↑ lean mass, ↑ gastrocnemius weight, ↑ muscle fiber size, ↑ bone mineral density Muscle function: improved muscle fiber morphology	Diabetes: ↓ fasting blood glucose Other: improved gut microbiota profile; metabolite and protein normalization	Gut–muscle axis: Beneficial microbial shifts (↑ Lachnospiraceae; ↓ Parabacteroides, Akkermansia). Proteostasis/HSP: Protein expression normalization (↑ Trim54; ↓ Arl6ip5, SNX6). Other: Normalized 44 metabolites; regulated β-alanine/histidine metabolism, ABC transporters, Ca^2+^ signaling.	[35]
Umbelliferone	Natural coumarin in carrots, parsley, celery, cumin, and fennel	Animal and Cell	Mice: db/db diabetic mice vs. non-diabetic controls. Cells: C2C12 muscle cells exposed to high glucose (50 mM)	Mice: 10 mg/kg/day orally for 8 weeks. Cells: 10–20 μM for 4 days	Muscle strength: ↑ grip strength Muscle mass: ↑ lean mass Muscle function: restored Type I fibers; ↑ myotube formation; ↑ myogenic markers (MyoD, Myogenin, Myh2); improved muscle morphology	Diabetes: ↓ HbA1c Obesity: ↓ fat mass	AMPK–SIRT1–PGC-1α: ↑ AMPK, ↑ SIRT1, ↑ PGC-1α → enhanced mitochondrial biogenesis. Mitochondrial QC/dynamics: ↑ MFN1, ↑ OPA1; ↓ DRP1, ↓ FIS1 (restored fusion–fission balance). FOXO/UPS: ↓ MuRF1, ↓ Atrogin-1, ↓ FoxO3a (reduced proteolysis). Akt–mTOR/Myogenesis: ↑ MyoD, ↑ Myogenin, ↑ Myh2 (promoted differentiation).	[36]
Lespedeza bicolor extract (LBE)	Lespedeza bicolor	Animal	Male C57BL/6J mice; type 2 diabetes induced by high-fat diet + 2 low-dose STZ injections	Oral gavage: 100 mg/kg/day (low) or 250 mg/kg/day (high), 12 weeks	Muscle mass: ↑ muscle weight, ↑ muscle fiber size	Diabetes: ↓ fasting glucose, ↓ HbA1c, ↑ glucose tolerance, ↑ insulin signaling	AMPK–SIRT1–PGC-1α: Activated pathway → ↑ mitochondrial biogenesis, ↑ energy metabolism. Akt–mTOR/GLUT4 pathway: ↑ GLUT4 translocation → enhanced insulin signaling. Inflammation/Oxidative stress: ↓ TNF-α, ↓ IL-6, ↓ MCP-1, ↓ 4-HNE. FOXO/UPS: ↓ FoxO3a, ↓ Atrogin-1, ↓ MuRF1 → reduced muscle proteolysis.	[37]
L-arabinose	Plant sugar (from hemicellulose in plant cell walls)	Animal	db/db mice (leptin receptor-deficient, obese, type 2 diabetes with muscle atrophy tendency)	Diet with 5% L-arabinose for 16 weeks	Muscle mass: ↑ gastrocnemius weight Muscle function: ↑ myogenesis (via ↑ IRS1/GLUT4)	Diabetes: ↓ fasting glucose, ↓ HOMA-IR, ↑ glucose tolerance, ↑ insulin sensitivity Obesity: ↓ body weight, ↓ fat mass	AMPK–SIRT1–PGC-1α: ↑ PGC-1α and ↑ CPT1b → enhanced fatty acid oxidation. Akt–mTOR/Insulin signaling: ↑ GLUT4, ↑ IRS1 → improved muscle glucose uptake and myogenesis. Inflammation: ↓ TNF-α, ↓ IL-6, ↓ IL-1β, ↓ NF-κB. Other: ↓ gluconeogenesis (↓ PEPCK, ↓ G6Pase), ↑ glycolysis (↑ GK).	[38]
Brazilian green propolis	*Baccharis dracunculifolia* extract (Yamada Bee Company, Japan)	Animal and Cell	Mice: Db/m (control) vs. Db/Db (sarcopenic obesity model) ± propolis. Cells: C2C12 myotubes exposed to palmitic acid ± propolis, artepillin C, or kaempferide	Mice: 0.08%, 0.4%, 2% propolis in chow, 8 weeks Cells: Propolis 100 µg/mL; Artepillin C 10.6 µg/mL; Kaempferide 1.89 µg/mL, 96 h	Muscle strength: ↑ grip strength Muscle mass: ↑ soleus mass, ↑ plantaris mass Muscle function: ↓ muscle atrophy genes; protection from PA-induced atrophy; ↑ ATP production; ↑ mitochondrial respiration	Diabetes: improved glucose tolerance Obesity: ↓ visceral fat Other: ↓ hepatic enzymes; ↓ liver fibrosis	Gut–muscle axis: Improved microbiota (↑ Bacteroidetes/Firmicutes ratio, ↑ Butyricicoccus, ↑ Acetivibrio); ↑ SCFAs. Inflammation/Immune modulation: ↓ inflammation; ↓ CD36; ↑ fatty acid excretion; macrophage shift to M2; ↑ ILC2. Mitochondrial function: Artepillin C and kaempferide protected mitochondria from palmitate, ↑ ATP, ↑ respiration. Other: CD36 suppression → reduced lipid uptake stress.	[29]
γ-Aminobutyric acid (GABA)	Four-carbon non-proteinogenic amino acid (abundant in vegetables, fruits, fermented foods)	Animal	Male C57BL/6J mice, young (3 mo) and aged (20 mo); HFD-induced obesity ± GABA	Oral gavage, 10 or 30 mg/kg/day, 8 weeks	Muscle strength: ↑ grip strength Muscle mass: ↑ lean mass, ↑ muscle fiber CSA Muscle function: ↓ Atrogin-1, ↓ MuRF1, ↓ myostatin; ↑ myogenic proteins (MyoD, Myogenin, Myf5/6)	Diabetes: ↓ fasting glucose, ↓ insulin resistance Obesity: ↓ visceral adipose tissue (VAT), ↓ adipocyte size Lipids: ↓ TG, ↓ TC, ↓ LDL; ↑ HDL Other: ↑ mitochondrial markers in muscle; ↑ thermogenic markers in fat; ↑ testosterone (aged mice)	Akt–mTOR anabolic signaling (PI3K/Akt activation, mTOR/4EBP1 ↑ → protein synthesis). FOXO/UPS proteolysis suppression (↓ FoxO3a, ↓ Atrogin-1, ↓ MuRF1). AMPK–SIRT1–PGC-1α activation (↑ PGC-1α, ↑ NRF1, ↑ TFAM, ↑ UCP3). Anti-inflammatory and antioxidant pathways (↓ TNF-α, ↓ IL-6, ↓ IL-1β). Other: Enhanced lipolysis/thermogenesis (p-PKA, ATGL, HSL, MGL; UCP1 in VAT)	[39]
Gintonin-enriched fraction (GEF)	Non-saponin glyco-lipoprotein fraction from Korean ginseng (Panax ginseng Meyer)	Animal	Male ICR mice (4 wk), divided into ND, HFD, HFD + GEF 50 mg/kg/day, HFD + GEF 150 mg/kg/day	Oral gavage, 50 mg/kg/day or 150 mg/kg/day for 6 weeks	Muscle strength: ↑ grip strength Muscle mass: ↑ skeletal muscle mass, ↑ muscle fiber CSA Muscle function: ↑ myogenic proteins (MyoD, Myogenin, MEF-2, MYH7); ↓ Atrogin-1, ↓ MuRF1	Obesity: ↓ body mass gain; ↓ visceral and subcutaneous WAT; ↓ adipocyte size Other: ↑ rectal temperature; ↑ mitochondrial biogenesis markers in muscle; WAT browning/thermogenesis	AMPK–SIRT1–PGC-1α activation (↑ PGC-1α, ↑ NRF1, ↑ TFAM, ↑ UCP3). Akt–mTOR anabolic signaling (↑ MyoD, ↑ Myogenin, ↑ MEF-2, ↑ MYH7 → myogenesis). FOXO/UPS proteolysis suppression (↓ Atrogin-1, ↓ MuRF1). Mitochondrial quality control/mitophagy (↑ mitochondrial biogenesis and energy expenditure—best fit under mitochondrial QC). Other: WAT browning and thermogenesis via AMPK → ↑ PKA/ATGL/HSL/MGL, ↑ PRDM16, ↑ UCP1	[40]
Panax ginseng berry extract (GBE)	Ginseng berries (Panax ginseng Meyer), standardized to 5% ginsenoside Re	Animal	Male C57BL/6 mice, HFD 9 weeks to induce sarcopenic obesity; then GBE 50, 100, 200 mg/kg/day groups	Oral gavage, 4 weeks with continued HFD	Muscle strength: ↑ grip strength Muscle mass: ↑ quadriceps mass, ↑ gastrocnemius mass, ↑ soleus mass, ↑ muscle fiber CSA	Diabetes: ↓ insulin Lipids: ↓ TG, ↓ TC, ↓ LDL; ↑ HDL/TC ratio Obesity: ↓ adipose tissue mass, ↓ adipocyte size; ↓ food efficiency ratio	Akt–mTOR anabolic signaling (restored IRS1–PI3K–Akt; ↑ mTOR, ↑ S6K1, ↑ 4E-BP1 → ↑ protein synthesis). FOXO/UPS proteolysis suppression (↓ FoxO3a, ↓ Atrogin-1, ↓ MuRF1). Anti-inflammatory and antioxidant pathways (↓ TNF-α, ↓ IL-6, ↓ IL-1β). Other: Inactivation of PKCθ/PKCζ	[41]
Codonopsis lanceolata (CL) extract	Perennial medicinal plant (roots; Jeonju, Korea).	Animal and Cell	Mice: Male C57BL/6, HFD 9 wks → obesity, then CL 6 wks with HFD. Groups: Normal, Control, CL50, CL100, CL200. Cells: C2C12 myotubes + palmitic acid ± TS	Mice: CL oral gavage 50, 100, 200 mg/kg/day for 6 weeks. Cells: TS 0.3 or 0.9 µg/mL, 24 h	Muscle strength: ↑ grip strength Muscle mass: ↑ gastrocnemius mass, ↑ quadriceps mass, ↑ soleus mass, ↑ muscle fiber CSA Muscle function: TS prevented PA-induced atrophy in C2C12 myotubes	Obesity/Diabetes/Other metabolic outcomes Obesity: ↓ body weight, ↓ fat mass, ↓ adipocyte CSA Lipids: ↓ TC, ↓ LDL, ↑ HDL/TC ratio Diabetes: ↓ intramyocellular TGs; improved insulin sensitivity (↓ HOMA-IR) Other: —	Akt–mTOR anabolic signaling (restored PI3K/Akt; ↑ p-S6K1, ↑ p-4EBP1 → ↑ protein synthesis). FOXO/UPS proteolysis suppression (↓ MuRF1, ↓ Atrogin-1 via ↑ p-FoxO3a). Other: Improved lipid metabolism in muscle (↓ SREBP-1c, DGAT2, SCD1; ↑ CPT1, ACOX1, UCP3)	[42]
Setaria viridis (SV) ethanol extract	Common annual grass native to Eurasia and North Africa (*Poaceae* family)	Animal	Male C57BL/6J mice (4 wk), 3 groups: ND (5% kcal fat), HFD (60% fat), HFD + SV (0.3% SV extract)	0.3% SV extract in HFD diet (~420 mg/kg/day), 20 weeks	Muscle mass: ↑ gastrocnemius (GAS) mass, ↑ quadriceps (QUA) mass, ↑ tibialis anterior (TA) mass, ↑ muscle fiber CSA, ↑ leg thickness Muscle function: ↑ IGF-1, ↓ myostatin	Obesity: ↓ body weight, ↓ WAT, ↓ intramuscular fat Lipids: ↓ TG, ↓ TC, ↓ non-HDL, ↑ HDL/TC ratio, ↓ ApoB/ApoA1 Diabetes: improved liver enzymes (ALT/AST) Other: ↓ inflammation, ↓ oxidative stress (↓ TBARS; ↑ GSH/GR/GPx), ↓ fibrosis	AMPK–SIRT1–PGC-1α activation (↑ AMPK, ↑ SIRT1, ↑ PGC-1α; docking: luteolin-7-O-glucoside binds AMPK ATP site → direct activator). Akt–mTOR anabolic signaling (↑ mTOR, ↑ p-S6K1, ↑ 4EBP1). FOXO/UPS proteolysis suppression (↓ Atrogin-1, ↓ MuRF1, ↓ FoxO3a). Mitochondrial quality control/mitophagy (↑ mitochondrial biogenesis). Anti-inflammatory and antioxidant pathways (↓ TNF-α, IL-6, IL-1β; ↓ TBARS; ↑ GSH/GR/GPx). Other: Improved lipid metabolism and fibrosis reduction	[43]
*Lonicera caerulea*	Honeysuckle berry, HB extract	Animal	Male C57BL/6 mice (6 wk) fed HFD (45% kcal fat); 6 groups: ND, HFD, HFD + Orlistat (20 mg/kg), HFD with different concentrations of HB extract	Oral gavage, 100, 200, and 400 mg/kg/day, 8 weeks	Muscle strength: ↑ grip strength Muscle mass: ↑ hindlimb muscle volume, ↑ muscle fiber CSA Muscle function: ↓ Atrogin-1, ↓ MuRF1; ↑ SIRT1, ↑ PGC-1α; ↑ antioxidant enzymes	Obesity: ↓ body weight gain, ↓ abdominal fat, ↓ subcutaneous fat (micro-CT) Lipids: ↓ serum TG, ↓ TC Other: ↓ leptin, ↑ adiponectin	AMPK–SIRT1–PGC-1α activation (↑ PGC-1α, ↑ SIRT1). FOXO/UPS proteolysis suppression (↓ Atrogin-1, ↓ MuRF1). Anti-inflammatory and antioxidant pathways (↑ SOD, ↑ GPx, ↑ CAT—antioxidant defense).	[44]
Resveratrol (RSV)	Natural polyphenol (grapes, berries, peanuts)	Animal and Cell	Rats: Young (3 mo) vs. aged (18 mo) Sprague–Dawley; HFD-fed aged rats ± RSV. Cells: L6 myotubes ± palmitate (0.75 mM) ± RSV (1–25 μM)	Rats: RSV 0.4% diet (~400 mg/kg/day), 10–20 wks. Cells: RSV 25 μM, 24 h	Muscle strength: ↑ grip strength Muscle mass: ↑ gastrocnemius (GA) mass, ↑ tibialis anterior (TA) mass, ↑ muscle fiber CSA Muscle function: Protected C2C12 myotubes from PA-induced atrophy (prevented ↓ myotube diameter, ↓ MHC); ↓ ROS; ↓ intracellular TG accumulation.	Obesity: ↓ body fat, ↓ intramuscular TG Lipids: ↓ serum TG, ↓ TC, ↓ LDL-C; ↑ HDL-C Other: ↑ mitochondrial function	AMPK–SIRT1–PGC-1α activation (PKA/LKB1 → AMPK activation; ↑ PGC-1α, ↑ TFAM, ↑ mtDNA). Mitochondrial quality control/mitophagy (improved mitochondrial biogenesis; ↑ MFN2, ↓ DRP1; ↑ ATP; ↓ ROS). Akt–mTOR anabolic signaling (↑ p-mTOR, ↑ p-S6K → restored protein synthesis). FOXO/UPS proteolysis suppression (↓ FoxO3a, ↓ Atrogin-1, ↓ MuRF1). Anti-inflammatory and antioxidant pathways (↓ ROS in PA-treated myotubes). Other: Upstream activation via PKA/LKB1	[28]
Resveratrol (RSV)	Natural polyphenol (grape, berries, peanuts)	Animal + in silico	Male C57BL/6 mice, 18 mo (aged). Groups: Control diet, HFD (sarcopenic obesity), HFD + RSV	0.4% RSV mixed in HFD (~400 mg/kg/day), 20 weeks	Muscle strength: ↑ grip strength Muscle mass: ↑ TA, GAS, QF muscle mass; ↑ muscle fiber CSA Muscle function: Improved muscle morphology; ↓ intramuscular lipid droplets; ↓ Atrogin-1 and MuRF1	Obesity: ↓ body weight; ↓ visceral and subcutaneous fat Lipids: ↓ TG, ↓ TC, ↓ LDL; ↑ HDL Diabetes: ↓ fasting glucose, ↓ insulin, ↓ HOMA-IR	FOXO/UPS proteolysis suppression (↓ Atrogin-1, ↓ MuRF1). Anti-inflammatory and antioxidant pathways (↓ IL-6, ↓ TNF-α, ↓ CRP). Other: Improved lipid and glucose metabolism (better insulin sensitivity).	[45]
Aged black garlic (ABG) and aged black elephant garlic (ABEG)	*Allium sativum* L., *Allium ampeloprasum* L. (aged extracts)	Animal and Cells	Male C57BL/6 mice, HFD-induced obesity (10 wks). Groups: ND, HFD, HFD + ABG, HFD + ABEG. Cells: C2C12 myotubes ± palmitate ± ABG/ABEG; 3T3-L1 adipocytes ± ABG/ABEG	Mice: oral gavage, 100 mg/kg/day for 10 weeks. Cells: ABG/ABEG for 48 h (C2C12) or 4 days (3T3-L1)	Muscle mass: ↑ muscle mass/body weight; ↑ MyHC; protection against PA-induced atrophy (C2C12) Muscle function: ↑ PGC-1α; restored mitochondrial markers (NRF1, TFAM); ↓ Atrogin-1 and MuRF1; ↑ myogenic factors (MyoD, Myogenin, MRF4)	Obesity: ↓ body weight gain; ↓ WAT mass; ↓ adipocyte size Lipids: ↓ TG, ↓ TC, ↓ LDL Other: ↓ ALT, ↓ AST	↑Akt–mTOR anabolic signaling ↑ protein synthesis) FOXO/UPS proteolysis suppression (↓ Atrogin-1, ↓ MuRF1). Mitochondrial quality control/mitophagy (↑ NRF1, ↑ TFAM, improved mitochondrial integrity). Akt–mTOR myogenesis (↑ MyoD, ↑ Myogenin, ↑ MRF4). Other: WAT browning/thermogenesis (↑ UCP1, ↑ PGC-1α); identification of S-methyl-L-cysteine and L-proline as bioactives.	[46]
Curcumin	Natural polyphenol	Animal (in vivo)	Rats: Male Sprague–Dawley, 5 weeks old. Groups: (1) Control, (2) Sham, (3) KOA model via Hulth surgery, (4) KOA + Curcumin	150 mg/kg/day orally (0.5% CMC vehicle) for 5 weeks	Muscle strength: Improved gait performance (↑ contact area, ↑ intensity, ↑ stance time) Muscle mass: ↑ quadriceps fiber CSA Muscle function: ↓ Atrogin-1 and MuRF1 (reduced proteolysis)	Other: ↓ ROS, ↑ SOD2 activity, ↓ excessive autophagy; improved cartilage integrity (↓ OARSI score)	Anti-inflammatory and antioxidant pathways (SIRT3–SOD2 activation →↓ROS). FOXO/UPS proteolysis suppression (↓ Atrogin-1, ↓ MuRF1). Proteostasis/HSP pathways (restored proteostasis; reduced excessive autophagy).	[47]
Plant-based polyphenol-rich protein (PRP)	Extracted from plant sources (rich in flavonoids and phenolic amino acids)	Animal	Aged mice (20 months old, sarcopenic model)	0.5% or 1% PRP in diet for 8 weeks	Muscle strength: ↑ grip strength Muscle mass: ↑ muscle fiber CSA, ↑ lean mass	Other: ↓ IL-6, ↓ TNF-α; ↑ mitochondrial activity	↑AMPK–SIRT1–PGC-1α activation ↑Akt–mTOR anabolic signaling → enhanced anabolic drive Anti-inflammatory and antioxidant pathways (↓ IL-6, ↓ TNF-α). Gut–muscle axis/microbiota–SCFA signaling (↑ SCFA-producing bacteria: Akkermansia, Lactobacillus)	[48]
Coix Seed Oil (CSO)	Coix lacryma-jobi L. (plant oil rich in linoleic acid and indirubin)	Animal and cell	Collagen-induced arthritis rats (RS model); C2C12 myotubes (Leptin-induced atrophy)	Rats: 2.1 or 8.4 g/kg oral for 28 days; Cells: 1000–2000 µg/mL 48 h	Muscle strength: ↑ grip strength Muscle mass: ↑ total muscle mass, ↑ muscle fiber CSA Muscle function: ↑ myogenic markers (MyoD, MyoG), ↓ atrophy markers (Atrogin-1, MuRF1)	Other: ↓ IL-6, ↓ TNF-α, ↓ leptin	FOXO/UPS proteolysis suppression (↓ Atrogin-1, ↓ MuRF1). Anti-inflammatory pathways (↓ IL-6, ↓ TNF-α, ↓ leptin). Gut–muscle axis/microbiota–SCFA signaling (↑ Lactobacillus, ↓ Bacteroides). Other: Suppressed leptin–JAK2–STAT3 axis; enhanced myogenesis and mitochondrial function.	[49]
Cinnamic acid derivatives (cinnamoylglycine, 4-methoxycinnamic acid, 3,4,5-trimethoxycinnamic acid, sinapinic acid)	Plant polyphenol metabolites produced via gut microbiota	Human clinical (cross-sectional metabolomics)	15 older adults (77–90 y), low vs. normal HGS	N/A: measured serum and fecal levels	Muscle strength: ↓ handgrip strength associated with ↓ circulating cinnamic acid metabolites	Other: ↓ fecal polyphenol metabolites linked with gut dysbiosis and impaired microbial metabolism	Anti-inflammatory and antioxidant pathways: Lower metabolites associated with ↑ inflammation. Gut–muscle axis/microbiota–SCFA signaling: Gut dysbiosis reduces microbial polyphenol metabolism → reduced beneficial cinnamic derivatives → poorer muscle function. Other: Decreased metabolites suggest ↓ mitochondrial efficiency and impaired cellular energetics.	[50]
Pea protein isolate supplementation	Pea protein isolate (NUTRALYS S85 Plus N; Roquette Frères)	Randomized, double-blind, controlled human trial	Healthy older males (72 ± 4 y); n = 31 assigned to whey (n = 10), pea (n = 11), collagen (n = 10)	Phase 1 (7 days): controlled diet providing protein at the RDA (0.8 g/kg BW/day). Phase 2 (next 7 days): same diet + pea protein supplement (25 g at breakfast + 25 g at lunch; total = 50 g/day).	Muscle mass: ↑ myofibrillar protein synthesis (~9%) Muscle function: ↑ mTORC1 and rpS6 activation; ↑ post-meal aminoacidemia (especially leucine); no change in physical activity.	Other: Pea protein matched whey in stimulating MPS; collagen showed no improvement	Akt–mTOR anabolic signaling (↑ mTORC1, ↑ rpS6 → ↑ MPS). Proteostasis/HSP pathways: (enhanced daily MPS overcoming anabolic resistance). Other: Improved amino acid distribution at low-protein meals enhanced total daily anabolic response.	[51]

**Table 2 nutrients-17-03656-t002:** Overview of Animal and Marine Source Interventions for Sarcopenic Obesity and Diabetic Sarcopenia.

Intervention	Source	Study Type	Population/Model	Dose and Duration	Muscle Outcomes	Metabolic Outcomes	Mechanism	Ref.
Dried whole egg; FMT from egg-fed donors	Whole chicken egg (dried)	Animal	Male db/db mice; Egg− vs. Egg+. Antibiotic-depleted mice for FMT(Egg-fed) vs. FMT(Control)	Egg: 1% egg in chow for 8 weeks FMT: Antibiotic pretreatment (age 6–8 wks), followed by FMT twice weekly until 16 weeks of age	Muscle strength: ↑ grip strength Muscle mass: ↑ soleus mass, ↑ plantaris mass, ↑ muscle fiber CSA Muscle function: ↑ differentiation markers (MyoD, Myogenin, MHC)	Obesity: ↓ visceral fat, ↑ spontaneous activity Diabetes: Improved glucose tolerance, ↑ insulin sensitivity Other: FMT recipients reproduced muscle and metabolic benefits → gut-mediated mechanism	Akt–mTOR anabolic signaling (↑ p70S6K, ↑ 4EBP1 → ↑ protein synthesis). FOXO/UPS proteolysis suppression (↓ AMPKα, ↓ FoxO1, ↓ MuRF1 → ↓ protein breakdown). Gut–muscle axis/microbiota–SCFA signaling (microbiota shift ↑ Vampirovibrio; FMT reproduced muscle and metabolic benefits). Other: ↑ intestinal amino acid transporters (Slc6a18/6a19/38a6) → ↑ circulating BCAAs and lysine; enhanced muscle amino acid availability.	[30]
Whey peptide (WP) ± resistance exercise (RE)	Hydrolyzed whey protein	Animal	Male C57BL/6J mice (8 mo), HFD (60% fat, 8 wks) → sarcopenic obesity. Groups: CON (normal diet), OB (HFD), RE (HFD + exercise), WP (HFD + WP), WPE (HFD + WP + exercise).	oral gavage, 1500 mg/kg/day, 8 weeks ladder climbing, 5×/week, 8–10 reps, 10–20% body weight load, 8 weeks.	Muscle strength: ↑ (with RE; strongest in WPE) Muscle mass: WP preserved lean mass; ↑ fiber CSA; WPE ↑ gastrocnemius weight Muscle function: WPE improved muscle morphology.	Obesity: WP ↓ fat gain, ↓ adipocyte size; WPE strongest ↓ body fat and improved composition Other: Synergistic effect when WP combined with resistance exercise	AMPK–SIRT1–PGC-1α activation (in adipose: ↑ AMPK/PGC-1α → improved energy metabolism). Akt–mTOR anabolic signaling (↑ Akt/mTOR in muscle → ↑ protein synthesis). FOXO/UPS proteolysis suppression (↓ Atrogin-1, ↓ MuRF1). Anti-apoptotic/antioxidant (↓ Bax; not a main category → placed under Other). Other: ↓ adipogenesis (↓ PPARγ, ↓ C/EBPα).	[52]
Whey protein + L-leucine + vitamin D	Whey protein supplement (18 g protein, 4.1 g leucine, 200 IU vit D3)	Human clinical trial (open, uncontrolled, pilot)	16 post-menopausal obese women (age 50–70 y, BMI 31.7–44.1, HOMA-IR ≥ 2.5, sarcopenic obesity, sedentary)	45 days, LCD 1000 kcal/day + daily supplement (total protein ~1.38 g/kg/day, vit D ~600 IU/day)	Muscle strength: ↑ handgrip strength (15.3 → 20.1 kg) Muscle mass: Lean mass preserved (55%) Muscle function: ↑ physical function (SPPB 7.5 → 8.9)	Obesity: ↓ BMI (37.6 → 35.7), ↓ waist circumference (107 → 102 cm) Diabetes: ↓ insulin (17.4 → 10.4 µIU/mL), ↓ HOMA-IR (4.8 → 2.3) Other: Mild ↑ BUN; overall well tolerated	Akt–mTOR anabolic signaling (leucine + whey → ↑ mTORC1 → ↑ muscle protein synthesis). Proteostasis/HSP pathways (preserved MPS during caloric restriction = maintained proteostasis). Other: Vitamin D support for muscle and bone; high-quality whey amino acids improving overall anabolic response.	[53]
Sarcomeal^®^ sachet (whey protein, creatine, glutamine, BCAAs, HMB) + vitamin D3	Commercial supplement (Karen Pharm., Iran)	Human RCT	60 adults with type 2 diabetes and sarcopenia (age 50–75 y, Tehran)	1 sachet/day (20 g protein, 2 g glutamine, 1.5 g creatine, 2 g HMB, 2 g BCAAs) + 1000 IU vit D daily; 12 weeks	Muscle strength: ↑ grip strength (+1.3 kg) Muscle mass: ↑ lean mass (+1.70 kg), ↑ lean mass index (LMI), ↑ skeletal muscle index (SMI) Muscle function: Preserved functional quality of life (QoL) vs. decline in controls	Obesity: Body weight maintained (no excess fat gain)	Akt–mTOR anabolic signaling (whey protein + leucine/BCAAs → ↑ MPS). FOXO/UPS proteolysis suppression (HMB + glutamine → ↓ muscle breakdown). Other: Creatine → ↑ muscle energy; vitamin D → supported muscle and bone function.	[27]
Krill oil supplementation	Antarctic krill oil	Animal (aging mice)	C57BL/6 mice (aging model)	25 g krill oil per kg chow (≈100–300 mg/kg body weight EPA/DHA) for 4 weeks	Muscle strength: ↑ grip strength; ↑ twitch force; ↑ tetanic force (EDL muscle) Muscle mass: No loss of muscle mass Muscle function: Preserved mitochondrial Ca^2+^ uptake; maintained excitation–contraction coupling	Other: Improved mitochondrial integrity and muscle contractile function	Akt–mTOR anabolic signaling (EPA/DHA → ↑ mTOR–p70S6K → ↑ muscle protein synthesis). Mitochondrial quality control/mitophagy (↑ Mfn2; improved mitochondrial dynamics & Ca^2+^ handling). Anti-inflammatory and antioxidant pathways (EPA/DHA reducing oxidative stress → preserved proteostasis). Other: ↑ MCU expression → improved mitochondrial Ca^2+^ homeostasis; cognitive improvements.	[54]
Omega-3 polyunsaturated fatty acids (PUFAs)	Dietary fish and seafood (EPA, DHA, DPA) + serum phospholipid omega-3 levels	Human	185 women with polycystic ovary syndrome (PCOS) (mean age 29 ± 6 y, BMI ≈ 21 kg/m^2^))	Dietary intake ≈ 1.05 ± 0.42 g/day total *n*-3 PUFAs; long-chain ≈ 44.3 ± 17.1 mg/day (EPA ≈ 19.8 mg, DHA ≈ 18.1 mg). Serum total *n*-3 ≈ 4.84 ± 1.96% of total phospholipid FAs	Muscle mass: ↑ muscle mass (DXA/BIA)	Diabetes: Lower HOMA-IR (β −0.09 to −0.18) Obesity: ↓ fat mass; ↓ body-fat %	Anti-inflammatory and antioxidant pathways (↓ NF-κB activity). Mitochondrial quality control/lipid oxidation (↑ mitochondrial β-oxidation). Other: Improved insulin signaling (↑ GLUT4, ↑ IRS-1); ↓ ER stress; activation of GPR120 pathway.	[55]

**Table 3 nutrients-17-03656-t003:** Overview of Microorganisms and Fermented Products Intervention Targeting Sarcopenic Obesity and Diabetic Sarcopenia.

Intervention	Source	Study Type	Population/Model	Dose and Duration	Muscle Outcomes	Metabolic Outcomes	Mechanism	Ref.
Young-donor gut microbiota (FMT)	Restored gut microbial community (youth-type microbiota)	Animal study	Aged C57BL/6J mice (22 months)	Oral FMT 3× per week × 8 weeks	Muscle strength: ↑ grip strength Muscle mass: ↑ lean mass Muscle function: ↑ mitochondrial biogenesis; ↓ muscle atrophy genes (MuRF1, Atrogin-1)	Other: ↓ inflammatory markers; rejuvenated gut microbial profile	AMPK–SIRT1–PGC-1α activation (↑ mitochondrial biogenesis). FOXO/UPS proteolysis suppression (↓ MuRF1, ↓ Atrogin-1). Gut–muscle axis/microbiota–SCFA signaling (↑ SCFA-producing bacteria: Lachnospiraceae, Akkermansia). Anti-inflammatory pathways (reduced systemic and muscle inflammation).	[56]
Lactobacillus paracasei P62, Bifidobacterium bifidum P61 (alone or combined)	Human gut isolates; probiotic strains	Animal + cell	C57. BL/6 aged mice (18 mo); C2C12 cells.	1 × 10^9^ CFU/mouse/day × 8 weeks 1 × 10^4^ CFU/mL in vitro	Muscle strength: ↑ grip strength Muscle mass: ↑ muscle mass Muscle function: ↑ running endurance; ↑ mitochondrial biogenesis markers (PGC-1α, SIRT1); ↑ MyHC; ↓ MuRF1, ↓ FOXO3a	Other: ↓ systemic IL-6, ↓ NF-κB (improved metabolic inflammation); restored gut microbiota composition (↑ Akkermansiaceae, ↓ Deferribacteraceae)	Akt–mTOR anabolic signaling (↑ Akt → ↑ protein synthesis). FOXO/UPS proteolysis suppression (↓ FOXO3a, ↓ MuRF1, ↓ Atrogin-1/MAFbx). Mitochondrial quality control/biogenesis (↑ PGC-1α, ↑ SIRT1). Anti-inflammatory pathways (↓ NF-κB, ↓ IL-6). Gut–muscle axis/microbiota–SCFA signaling (restored beneficial taxa such as Akkermansiaceae).	[57]
Bifidobacterium pseudolongum, Turicibacter sanguinis, Clostridium cocleatum	Gut-derived bacterial isolates	Animal + Cell study	Aged C57BL/6 mice (18–24 mo); C2C12 myotubes	Oral gavage 1 × 10^9^ CFU/day × 6 weeks; metabolites 48 h in vitro	Muscle strength: ↑ grip strength Muscle mass: ↑ muscle fiber CSA Muscle function: ↑ mitochondrial biogenesis markers (PGC-1α, SIRT1)	Diabetes: ↑ insulin sensitivity Other: ↓ IL-6, ↓ TNF-α; improved systemic metabolic inflammation and gut ecosystem	AMPK–SIRT1–PGC-1α activation (↑ mitochondrial biogenesis). FOXO/UPS proteolysis suppression (suppressed FOXO3a). Anti-inflammatory pathways (↓ NF-κB, ↓ IL-6, ↓ TNF-α). Gut–muscle axis/microbiota–SCFA signaling (bacteria-derived metabolites improved insulin sensitivity and muscle metabolism).	[58]
Gut microbiota composition and loss of short-chain fatty acid–producing bacteria	Human gut microbiota	Clinical observational	Older adults ≥ 60 y (27 sarcopenic/possibly sarcopenic vs. 60 controls)	Single-time fecal sampling analyzed by 16S rRNA sequencing	Muscle strength: ↓ grip strength Muscle mass: ↓ muscle mass	Diabetes: Insulin resistance associated with ↑ LPS biosynthesis Other: ↓ microbial diversity; ↓ butyrate-producing genera (Lachnospira, Fusicantenibacter, Roseburia, Eubacterium, Lachnoclostridium); ↑ Lactobacillus; ↑ systemic inflammation	Gut–muscle axis/microbiota–SCFA signaling (↓ SCFA producers → ↓ butyrate → impaired muscle mitochondrial energy metabolism). Anti-inflammatory pathways (↑ LPS biosynthesis → systemic inflammation). Other: ↓ amino acid biosynthesis pathways; ↓ transporters/cytoskeletal proteins contributing to muscle degradation.	[59]
Lacticaseibacillus paracasei PS23 (Lactica™)	Probiotic strain (fermented food bacterium)	Animal and cell study	20-month-old C57BL/6J mice; C2C12 myotubes	1 × 10^9^ CFU/day orally × 12 weeks; metabolites 48 h	Muscle strength: ↑ grip strength Muscle mass: ↑ muscle fiber CSA Muscle function: ↑ PGC-1α expression; ↑ AMPK phosphorylation	Diabetes: ↑ glucose tolerance; improved insulin sensitivity Other: ↓ systemic inflammatory cytokines (IL-6, TNF-α); improved gut microbiota composition	AMPK–SIRT1–PGC-1α activation (↑ AMPK phosphorylation, ↑ PGC-1α → enhanced mitochondrial biogenesis). Anti-inflammatory pathways (↓ IL-6, ↓ TNF-α). Gut–muscle axis/microbiota–SCFA signaling (improved gut microbial profile contributing to metabolic benefits).	[60]
Bovine colostrum-derived exosomes (BCEs)	Milk exosome fraction from colostrum	Animal and cell	C2C12 cells; C. elegans PD4251; DEX-treated C57BL/6J mice	C2C12 cells; C. elegans PD4251; DEX-treated C57BL/6J mice	Muscle strength: ↑ grip strength Muscle mass: ↑ myotube diameter Muscle function: ↑ MyoD, ↑ Myogenin (enhanced myogenesis); ↓ MuRF1 & ↓ Atrogin-1 (reduced proteolysis)	Other: ↓ TNF-α (reduced metabolic inflammation); ↑ Lachnospiraceae; restored L-alanine and succinic acid → improved gut–muscle metabolic axis	FOXO/UPS proteolysis suppression (↓ FoxO3a activity; ↓ MuRF1/Atrogin-1). Anti-inflammatory pathways (↓ TNF-α). Gut–muscle axis/microbiota–SCFA signaling (↑ Lachnospiraceae; restored gut-derived metabolites).	[61]

**Table 4 nutrients-17-03656-t004:** Overview of Environmental and Stress Modulation Interventions for Sarcopenic Obesity and Diabetic Sarcopenia.

Intervention	Source	Study Type	Population/Model	Dose and Duration	Muscle Outcomes	Metabolic Outcomes	Mechanism	Ref.
Exercise training (aerobic, resistance)	-	Animal and Human	Diabetic rodents; obese/sarcopenic elderly	5–14 weeks of training	Muscle function: ↑ antioxidant enzymes (SOD, CAT); ↓ MDA; improved mitochondrial function; cardioprotection	Other: Improved systemic redox status and mitochondrial resilience	Mitochondrial quality control/mitophagy: Enhanced CASA/autophagy, improved mitochondrial turnover. Proteostasis/HSP pathways: ↑ HSP70, HSP27, CRYAB → improved protein folding and aggregation control. Anti-inflammatory and antioxidant: ↑ SOD/CAT; ↓ lipid peroxidation. Other: Restores global muscle proteostasis and resilience.	[62]
Thermal manipulation (TM) during incubation	Environmental incubator temperature	Animal	Fertilized *Ross 308* broiler eggs	39.5 °C, relative humidity 65%, applied from ED13–ED17; exposure increased 2 → 10 h/day; embryos collected ED18	-	Other: ↓ HSP70/HSP90 expression in embryonic brain; presence of histopathological lesions (gliosis, edema)	Proteostasis/HSP pathways: Suppressed heat shock response (↓ HSP70/HSP90) during embryogenesis. Other: Indicates impaired cellular stress resilience and developmental adaptation under TM.	[63]
Melatonin and Exercise	Melatonin (Sigma-Aldrich, drinking water) and treadmill training	Animal and Cell	Mice: Male SAMP8, 24 weeks old, HFD to induce sarcopenic obesity. Groups: ND, HFD-control, HFD + melatonin, HFD + exercise, HFD + melatonin + exercise. Cells: Primary myoblasts from satellite cells and C2C12 myoblasts exposed to H_2_O_2_ ± melatonin	Mice: Melatonin ~10 mg/kg/day in drinking water, 8 weeks. Exercise: treadmill 30 min/day, 5 days/week, 8 weeks. Cells: Melatonin treatment during senescence assays	Muscle strength: ↑ grip strength Muscle mass: ↑ gastrocnemius mass, ↑ tibialis anterior mass, preserved soleus CSA Muscle function: ↓ extramyocyte space, preserved Pax7^+^ satellite cell pool, improved regenerative capacity Muscle function (cell level): ↓ SA-β-gal^+^ senescent cells, ↑ BrdU^+^ proliferation, ↑ MyoD^+^ commitment, ↑ fusion index, ↑ myotube diameter	Diabetes: ↓ fasting glucose	Mitochondrial quality control: ↑ mitochondrial calcium retention; protection from oxidative damage. Proteostasis/anti-senescence: ↓ p16^Ink^ → reduced senescence burden; normalization of cell cycle regulators. Akt–mTOR/myogenesis: ↑ MyoD, ↑ proliferation and differentiation, ↑ myotube fusion index. Anti-inflammatory and antioxidant: Melatonin reduced oxidative stress-induced senescence in vitro (H_2_O_2_ model). Other mechanisms: Maintenance of satellite cell function → enhanced muscle regeneration.	[64]

## Data Availability

No new data were created or analyzed in this study. Data sharing is not applicable to this article.

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
