# Peer review of "Bioactive Nutritional Components Within the Planetary Health Diet for Preventing Sarcopenic Obesity and Diabetic Sarcopenia: A Systematic Review"

_nutrients, 2025, doi:10.3390/nu17233656_

Round 1

Reviewer 1 Report

Comments and Suggestions for Authors

This review aims to examine the correlations of sarcopenic obesity and diabetic sarcopenia with dietary patterns especially within the planetary health diet framework. The conclusion is that certain foods improve sarcopenia and some of the mechanisms are described. The paper is well written with the studies presented in 5 tables which make the findings easy to find. The discussion is also clearly written. One missing topic that I not discussed is the effect of the GLP-1 agonists on muscle metabolism. However the paper is quite long so perhaps a small statement could be considered.  

Minor Points

  1. In table 1 on page 7 bottom lines (Codonopsis) and all those on page 8 plus on page 9 in the Sources column in the third entry (Allium) and in table 3 for 2 of the studies the entries are presented in Italics. There is nothing I could find in the paper to suggest why these facts are in italics. All of the other entries in the tables are normal print.
  2. Line 270 Figure X should be Figure 3  

Author Response

Response to Reviewer #1:

This review aims to examine the correlations of sarcopenic obesity and diabetic sarcopenia with dietary patterns especially within the planetary health diet framework. The conclusion is that certain foods improve sarcopenia and some of the mechanisms are described. The paper is well written with the studies presented in 5 tables which make the findings easy to find. The discussion is also clearly written. One missing topic that I not discussed is the effect of the GLP-1 agonists on muscle metabolism. However the paper is quite long so perhaps a small statement could be considered. 

Minor Points

  1. In table 1 on page 7 bottom lines (Codonopsis) and all those on page 8 plus on page 9 in the Sources column in the third entry (Allium) and in table 3 for 2 of the studies the entries are presented in Italics. There is nothing I could find in the paper to suggest why these facts are in italics. All of the other entries in the tables are normal print.
  2. Line 270 Figure X should be Figure 3

Response: Thank you for this thoughtful suggestion. The effects of GLP-1 receptor agonists on muscle metabolism are indeed an interesting and emerging area. However, other reviewers recommended removing pharmacological interventions to keep the scope tightly aligned with the aim of the review, which focuses specifically on dietary patterns and nutrition-derived bioactive components within the Planetary Health Diet framework. In light of this, we did not incorporate GLP-1 agonists or other drug-based mechanisms to maintain consistency with the revised scope. Additionally, we have corrected the formatting inconsistencies in Tables 1 and 3, ensuring that all entries now appear in a uniform font style. The citation for “Figure X” has also been revised to “Figure 3” as recommended.

Reviewer 2 Report

Comments and Suggestions for Authors

Review of ”Bioactive Nutritional Components within the Planetary Health Diet for Preventing Sarcopenic Obesity and Diabetic Sarcopenia” (nutrients-3972425)

This review article focuses on the effect of planetary health diet on sarcopenic obesity and diabetic sarcopenia. This study is potentially interesting; however, several problems should be solved.

Expression of outcomes and mechanisms are vague and difficult to understand in Tables. Outcome should be presented separately for sarcopenia (muscle strength, muscle mass, and physical activity), obesity (body weight, diabetes, lipids, and liver function), and other items. The mechanism should also be described based on the contents of Figure 2, 4, and 5.

Author Response

Response to Reviewer #2:

Review of ”Bioactive Nutritional Components within the Planetary Health Diet for Preventing Sarcopenic Obesity and Diabetic Sarcopenia” (nutrients-3972425)

This review article focuses on the effect of planetary health diet on sarcopenic obesity and diabetic sarcopenia. This study is potentially interesting; however, several problems should be solved.

Expression of outcomes and mechanisms are vague and difficult to understand in Tables. Outcome should be presented separately for sarcopenia (muscle strength, muscle mass, and physical activity), obesity (body weight, diabetes, lipids, and liver function), and other items. The mechanism should also be described based on the contents of Figure 2, 4, and 5.

Line 33: “SNP” must be defined

Response: We thank the reviewer for the valuable comments. Tables 1–4 have been thoroughly revised to incorporate these suggestions, with all outcome categories reorganized as recommended and the mechanistic descriptions refined to explicitly align with the pathways illustrated in Figures 2–4. We believe these revisions significantly improve the clarity, accuracy, and overall coherence of the presented information.

Reviewer 3 Report

Comments and Suggestions for Authors

The manuscript presents a timely and comprehensive review addressing the interrelationships between sarcopenia, sarcopenic obesity, and diabetic sarcopenia within the context of dietary patterns, particularly the Planetary Health Diet (PHD). The topic is of considerable relevance given the global rise in aging populations and the growing emphasis on sustainable nutrition. The paper is well-structured, with a clear description of the background and a commendable attempt to integrate nutritional and environmental perspectives.

However, several aspects could be refined to further strengthen the scientific rigor and coherence of the manuscript. Clarifying the conceptual link between the PHD and sarcopenia, defining key terms, and ensuring consistency with current diagnostic criteria would enhance the clarity and impact of the work.

Abstract

  1. Inconsistent study counts (Abstract/Results/PRISMA).
    The Abstract states 45 included studies, whereas the Results section reports 99. Figure 1 (PRISMA) also appears inconsistent. Please reconcile these numbers across the Abstract, Methods, Results, and PRISMA flow diagram, ensuring each screening stage (identified, deduplicated, excluded with reasons, finally included) is numerically consistent.

Introduction

  1. Clarification of the Rationale for Linking Sarcopenia and the Planetary Health Diet.

The rationale for linking sarcopenia and the Planetary Health Diet should be elaborated. The authors are encouraged to clarify how the PHD framework specifically contributes to addressing sarcopenic obesity or diabetic sarcopenia, beyond its general nutritional and sustainability benefits.

  1. Definition of Key Terms (SO and DS).

The terms “sarcopenic obesity (SO)” and “diabetic sarcopenia (DS)” are introduced in the last paragraph without clear definitions. It would strengthen the introduction to briefly define these conditions and cite the relevant diagnostic criteria.

  1. Alignment with Current Consensus Definitions.

The authors should consider citing recent consensus definitions such as EWGSOP2 (2019) or AWGS (2019) to align the description of sarcopenia with current international standards.

Materials and Methods

  1. Databases reported are inconsistent.
    The Abstract lists PubMed, Scopus, and Google, while the Results cite PubMed, ScienceDirect, Google Patent, and the Cochrane Library. For transparency and reproducibility, please report only the databases/search engines that were actually used, and use a single, consistent list throughout the manuscript (Abstract, Methods, and Results). In Section 2.3, you stated that filters were applied to restrict the results to human studies. However, several data sets presented in the manuscript are derived from animal and cell models. Please revise the Methods section accordingly to accurately reflect the actual inclusion criteria.

Results

  1. Scope drift toward pharmaceutical agents.
    The stated focus is “Bioactive Nutritional Components within the Planetary Health Diet,” yet the review includes several pharmaceutical agents (e.g., GLP-1 receptor agonists, STING inhibitors, BAM15, FGF19). This conflicts with a diet/nutrition-focused review. Please either ‘Remove pharmacologic agents and concentrate on foods, dietary patterns, and nutrition-derived bioactives’ or ‘Explicitly separate diet-based versus drug-based evidence and interpret them independently’
     Relatedly, much of the Results centers on isolated compounds/extracts/supplements/drugs rather than dietary patterns. Please include more studies that specifically examine dietary patterns and their relationships or interventions with SO and DS outcomes. If such studies are limited, please state this explicitly in the Discussion and describe it as an evidence gap.

  1. SO/DS case definition and model suitability must be explicit in tables and text.
    Many animal studies use models such as high-fat diet obesity, KOA, dexamethasone-induced atrophy, arthritis, or aging. It is difficult for readers to determine whether these meet SO and/or DS criteria. Please define the operational criteria for SO and DS used in this review (e.g., muscle mass, function plus adiposity, glycemic impairment thresholds). Indicate in the evidence tables whether each study meets SO/DS criteria (Yes/No/Unclear) and how. If a study does not meet criteria, mark it clearly and provide appropriate caveats in the Discussion.

Discussion

  1. Temper conclusions to the strength of evidence. Avoid over-extrapolation.
    Mechanistic summaries are helpful, but several statements appear to generalize from small, preclinical models to human clinical outcomes. Please revise to avoid over-interpretation: separate preclinical from clinical evidence in the narrative, moderate the language accordingly (e.g., “may,” “suggests,” “limited evidence”).

Overall

  1. Typographical and placeholder errors.
    Please correct residual placeholders and hyphenation artifacts. (e.g., Section 4.2, Line 321 “Figure X,” and Line 342 “com-prehensive” and run a thorough copyedit to eliminate similar issues throughout).

Reference formatting not compliant with journal style.
 Multiple references do not conform to the journal’s required style. Please reformat all citations and the reference list to match the journal’s guidelines.

Author Response

Response to Reviewer #3:

The manuscript presents a timely and comprehensive review addressing the interrelationships between sarcopenia, sarcopenic obesity, and diabetic sarcopenia within the context of dietary patterns, particularly the Planetary Health Diet (PHD). The topic is of considerable relevance given the global rise in aging populations and the growing emphasis on sustainable nutrition. The paper is well-structured, with a clear description of the background and a commendable attempt to integrate nutritional and environmental perspectives.

However, several aspects could be refined to further strengthen the scientific rigor and coherence of the manuscript. Clarifying the conceptual link between the PHD and sarcopenia, defining key terms, and ensuring consistency with current diagnostic criteria would enhance the clarity and impact of the work.

Abstract

Inconsistent study counts (Abstract/Results/PRISMA).

The Abstract states 45 included studies, whereas the Results section reports 99. Figure 1 (PRISMA) also appears inconsistent. Please reconcile these numbers across the Abstract, Methods, Results, and PRISMA flow diagram, ensuring each screening stage (identified, deduplicated, excluded with reasons, finally included) is numerically consistent.

Response: Thank you for pointing out the inconsistencies in the reported study counts. We have carefully reviewed and corrected the numbers across the Abstract, Methods, Results, and PRISMA flow diagram to ensure that all screening stages are now numerically consistent.

Introduction

  1. Clarification of the Rationale for Linking Sarcopenia and the Planetary Health Diet.

The rationale for linking sarcopenia and the Planetary Health Diet should be elaborated. The authors are encouraged to clarify how the PHD framework specifically contributes to addressing sarcopenic obesity or diabetic sarcopenia, beyond its general nutritional and sustainability benefits.

Response: Thank you for this helpful comment. We have added a brief clarification in the Introduction to better outline the rationale for connecting the Planetary Health Diet with sarcopenic obesity and diabetic sarcopenia. Specifically, we note that the nutrient composition of the PHD may assist in characterizing metabolic responses that are relevant to these conditions, such as those related to protein synthesis, inflammation, and oxidative stress

  1. Definition of Key Terms (SO and DS).

The terms “sarcopenic obesity (SO)” and “diabetic sarcopenia (DS)” are introduced in the last paragraph without clear definitions. It would strengthen the introduction to briefly define these conditions and cite the relevant diagnostic criteria.

Response: Thank you for this suggestion. We have added brief definitions of both sarcopenic obesity (SO) and diabetic sarcopenia (DS) in the Introduction.

  1. Alignment with Current Consensus Definitions.

The authors should consider citing recent consensus definitions such as EWGSOP2 (2019) or AWGS (2019) to align the description of sarcopenia with current international standards.

Response: Thank you for this important comment. We have updated the Introduction to include current consensus definitions and now cite the AWGS 2019 criteria to ensure alignment with internationally recognized standards for sarcopenia.

Materials and Methods

  1. Databases reported are inconsistent.

The Abstract lists PubMed, Scopus, and Google, while the Results cite PubMed, ScienceDirect, Google Patent, and the Cochrane Library. For transparency and reproducibility, please report only the databases/search engines that were actually used, and use a single, consistent list throughout the manuscript (Abstract, Methods, and Results). In Section 2.3, you stated that filters were applied to restrict the results to human studies. However, several data sets presented in the manuscript are derived from animal and cell models. Please revise the Methods section accordingly to accurately reflect the actual inclusion criteria.

Response: Thank you for pointing this out. We have corrected the inconsistency in reported databases: PubMed, Scopus, and Google are now the only sources listed, and this has been made fully consistent across the Abstract, Methods, and Results. We also appreciate the reviewer’s comment regarding the statement about “restricting results to human studies.” This was a wording mistake. Our search and inclusion criteria did include human, animal, and cell studies, and Section 2.3 has been revised to clearly reflect this.

Results

  1. Scope drift toward pharmaceutical agents.

The stated focus is “Bioactive Nutritional Components within the Planetary Health Diet,” yet the review includes several pharmaceutical agents (e.g., GLP-1 receptor agonists, STING inhibitors, BAM15, FGF19). This conflicts with a diet/nutrition-focused review. Please either ‘Remove pharmacologic agents and concentrate on foods, dietary patterns, and nutrition-derived bioactives’ or ‘Explicitly separate diet-based versus drug-based evidence and interpret them independently’

Response: Thank you for this important observation. To maintain alignment with the stated scope of our review, we have removed all pharmacological agents (e.g., GLP-1 receptor agonists, STING inhibitors, BAM15, FGF19) from the Results and Discussion sections. The revised manuscript now focuses exclusively on foods and nutrition-derived bioactives.

 Relatedly, much of the Results centers on isolated compounds/extracts/supplements/drugs rather than dietary patterns. Please include more studies that specifically examine dietary patterns and their relationships or interventions with SO and DS outcomes. If such studies are limited, please state this explicitly in the Discussion and describe it as an evidence gap.

Response: Thank you for this comment. Studies directly examining whole dietary patterns in relation to SO and DS are indeed limited. We have added a statement in the Results/Discussion to explicitly acknowledge this evidence gap and to clarify that most available studies focus on isolated compounds, extracts, or supplements rather than full dietary patterns.

  1. SO/DS case definition and model suitability must be explicit in tables and text.

Many animal studies use models such as high-fat diet obesity, KOA, dexamethasone-induced atrophy, arthritis, or aging. It is difficult for readers to determine whether these meet SO and/or DS criteria. Please define the operational criteria for SO and DS used in this review (e.g., muscle mass, function plus adiposity, glycemic impairment thresholds). Indicate in the evidence tables whether each study meets SO/DS criteria (Yes/No/Unclear) and how. If a study does not meet criteria, mark it clearly and provide appropriate caveats in the Discussion.

Response: Thank you for this helpful comment. We agree that SO and DS require clear definitions in human studies; however, formal diagnostic criteria (such as AWGS 2019) cannot be applied to animal or cell models. Most experimental models reflect broader metabolic disturbances—such as adiposity, insulin resistance, inflammation, or muscle atrophy—that overlap substantially between SO and DS. Because these mechanisms are closely intertwined, preclinical studies rarely distinguish SO from DS as separate entities, making a binary classification (Yes/No/Unclear) difficult to apply in a meaningful or standardized way.

Discussion

  1. Temper conclusions to the strength of evidence. Avoid over-extrapolation.

Mechanistic summaries are helpful, but several statements appear to generalize from small, preclinical models to human clinical outcomes. Please revise to avoid over-interpretation: separate preclinical from clinical evidence in the narrative, moderate the language accordingly (e.g., “may,” “suggests,” “limited evidence”).

Response: Thank you for this helpful comment. We have reviewed the Discussion and adjusted the language to better reflect the varying strengths of evidence. We have also added qualifying terms (e.g., “may,” “suggests,” “limited evidence”) where appropriate to avoid over-extrapolation and to ensure that conclusions remain consistent with the scope and quality of the available studies.

Overall

Typographical and placeholder errors.

Please correct residual placeholders and hyphenation artifacts. (e.g., Section 4.2, Line 321 “Figure X,” and Line 342 “com-prehensive” and run a thorough copyedit to eliminate similar issues throughout).

  1. Reference formatting not compliant with journal style.

Multiple references do not conform to the journal’s required style. Please reformat all citations and the reference list to match the journal’s guidelines.

Response: Thank you for pointing this out. We have reviewed the manuscript for placeholder text, hyphenation artefacts, and typographical inconsistencies, and have corrected the issues identified by the reviewer. The reference formatting has also been adjusted to better align with the journal’s required style.

Reviewer 4 Report

Comments and Suggestions for Authors

The text is scientifically sound, well-referenced, and mechanistically rich, making it suitable for a peer-reviewed nutrition/metabolism journal. Minor revisions to improve clarity, specificity, and structural flow will substantially enhance readability and impact.

ABSTRACT:

The authors must clearly state the objective of the study in the abstract.

Abstracts normally do not have a discussion section.

In the abstract, it is not necessary to include PROSPERO in the abstract.

INTRODUCTION

The last paragraph of the introduction should be removed or written in the methods section.

MATERIALS AND METHODS

Authors should try to ensure that figures are accompanied by their captions; that they are not separated by pages.

Given the large number of selected articles (99), the authors should find a more practical and simpler way to analyze them so that the table isn't so long. It can be a bit cumbersome.

DISCUSSION

Authors should start with their main finding.

Figure 2 doesn't seem to add any information to the text. It should be deleted.

Given the large number of articles to be analyzed in the discussion, the authors should be able to synthesize information so that the discussion does not become too lengthy and the reader doesn't get lost. They should NOT provide basic information or information that could be included in the introduction.

The authors repeat information already provided in the text. They should focus on justifying their results.

Authors must include a practical application section.

CONCLUSION

The conclusion should be 2-3 sentences explaining the findings of this study. It should be concise and clear. The conclusion should be better expressed; do not restate the results.

The conclusion does not reflect the findings of the study

The limitation section should go at the end of the discussion section

Author Response

Response to Reviewer #4:

The text is scientifically sound, well-referenced, and mechanistically rich, making it suitable for a peer-reviewed nutrition/metabolism journal. Minor revisions to improve clarity, specificity, and structural flow will substantially enhance readability and impact.

ABSTRACT: The authors must clearly state the objective of the study in the abstract. Abstracts normally do not have a discussion section. In the abstract, it is not necessary to include PROSPERO in the abstract.

Response: We thank the reviewer for this helpful comment. We have revised the abstract accordingly by clearly stating the objective of the study, removing the discussion-type content, and omitting the PROSPERO information as recommended.

INTRODUCTION: The last paragraph of the introduction should be removed or written in the methods section.

Response: We thank the reviewer for the comment. The last paragraph of the introduction has been revised and partially removed to ensure that methodological details are presented only in the Methods section.

MATERIALS AND METHODS: Authors should try to ensure that figures are accompanied by their captions; that they are not separated by pages. Given the large number of selected articles (99), the authors should find a more practical and simpler way to analyze them so that the table isn't so long. It can be a bit cumbersome.

Response: We sincerely thank the reviewer for these helpful suggestions. We have revised the layout so that figures now appear together with their captions and are no longer separated across pages. In addition, following Reviewer 2’s recommendation, we reorganized the tables by separating outcomes (sarcopenia-related, obesity/metabolic-related, and other outcomes) from mechanistic findings.

DISCUSSION: Authors should start with their main finding. Figure 2 doesn't seem to add any information to the text. It should be deleted. Given the large number of articles to be analyzed in the discussion, the authors should be able to synthesize information so that the discussion does not become too lengthy and the reader doesn't get lost. They should NOT provide basic information or information that could be included in the introduction. The authors repeat information already provided in the text. They should focus on justifying their results. Authors must include a practical application section.

Response: Thank you for the helpful comments. We have reorganized the Discussion to begin with the main findings and removed unnecessary background information. The section involving Figure 2 has been fully rewritten so that the figure now directly supports the revised content by summarizing the multifactorial risk factors relevant to our analysis, rather than repeating introductory material. We have also shortened overlapping text, focused the discussion on interpreting our results, and added a new “Practical Applications” subsection as recommended. We believe these revisions greatly improve clarity and readability.

CONCLUSION: The conclusion should be 2-3 sentences explaining the findings of this study. It should be concise and clear. The conclusion should be better expressed; do not restate the results. The conclusion does not reflect the findings of the study. The limitation section should go at the end of the discussion section.

Response: Thank you for this comment. The Conclusion section has been revised to be more concise and to clearly reflect the key findings of the review without repeating detailed results. In addition, the limitation statement has been placed at the end of the Discussion section as recommended.

Reviewer 5 Report

Comments and Suggestions for Authors

The manuscript is very well structured and presented and is of considerable interest. However, it is not clear how plant-based dietary patterns intersect with bioactive nutritional components. In other words, the difference in effectiveness in preventing sarcopenia between the various plant-based patterns in which these components can be found is unclear. It would be interesting, for example, to compare the aforementioned Mediterranean diet with a vegetarian and vegan diet or with other more typically Eastern dietary patterns, perhaps in a subsequent scientific study.

Author Response

Response to Reviewer #5:

The manuscript is very well structured and presented and is of considerable interest. However, it is not clear how plant-based dietary patterns intersect with bioactive nutritional components. In other words, the difference in effectiveness in preventing sarcopenia between the various plant-based patterns in which these components can be found is unclear. It would be interesting, for example, to compare the aforementioned Mediterranean diet with a vegetarian and vegan diet or with other more typically Eastern dietary patterns, perhaps in a subsequent scientific study.

Response: Thank you for this thoughtful comment. We agree that different plant-based dietary patterns may vary in their distribution of bioactive nutritional components, which could influence their potential effectiveness in preventing sarcopenia. Currently however, direct comparative studies examining specific dietary patterns, such as Mediterranean, vegetarian, vegan, or Eastern diets, in relation to sarcopenic obesity or diabetic sarcopenia are currently very limited. For this reason, it was not possible to draw pattern-specific conclusions within the scope of the present review. We have added a paragraph in the Discussion to acknowledge this evidence gap and to note that future research comparing these dietary patterns would be valuable.

Round 2

Reviewer 2 Report

Comments and Suggestions for Authors

For tables, I think it would be clearer to add columns for muscle strength, muscle mass, physical function, body weight, diabetes, and etc.

Author Response

Comments:

“For tables, I think it would be clearer to add columns for muscle strength, muscle mass, physical function, body weight, diabetes, and etc.”

Response:

Thank you for this helpful follow-up suggestion. To further improve clarity while avoiding excessively wide tables, we revised Tables 1–4 by merging sarcopenia-related and obesity/diabetes-related outcomes into a single column labelled “Metabolic Outcomes.” This consolidated column now lists the specific measures evaluated in each study (e.g., muscle strength, muscle mass, physical function, body weight, diabetes markers, lipid profiles, liver function). This approach presents all relevant outcomes clearly while maintaining a table layout that complies with the journal’s formatting constraints and prevents the readability issues that would arise from adding multiple separate columns. We appreciate your suggestion, which has resulted in a clearer and more practical organization of the evidence.

Reviewer 3 Report

Comments and Suggestions for Authors

The authors have addressed the previous comments thoroughly, and the manuscript has improved significantly in clarity and structure. Only a few additional refinements are needed to further enhance the overall quality of the paper.

1. In academic reviews, “Google Scholar” is typically used as a scientific search platform, whereas “Google” is not considered a formal database. Please confirm whether Google Scholar was actually used and revise the Methods section accordingly to ensure accuracy and reproducibility.

2. Typographical Error in Results (Line 230). In the Results section, “SO and DO” appears to be a typographical error and should likely be corrected to “SO and DS.”Please re-check the entire manuscript for typographical or spelling inconsistencies to ensure that no errors remain.

3. The Limitations should be expanded to explicitly address the following points.

‘Wide heterogeneity in dose and duration among interventions’ and ‘The limited proportion of human studies relative to animal or cell-based studies’

Adding these points will more accurately reflect the constraints of the current evidence base and improve the scientific rigor of the review.

Author Response

The authors have addressed the previous comments thoroughly, and the manuscript has improved significantly in clarity and structure. Only a few additional refinements are needed to further enhance the overall quality of the paper.

Comments 1: In academic reviews, “Google Scholar” is typically used as a scientific search platform, whereas “Google” is not considered a formal database. Please confirm whether Google Scholar was actually used and revise the Methods section accordingly to ensure accuracy and reproducibility.

Response 1:

Thank you for the clarification. We confirm that Google Scholar, not Google, was used as the supplementary search platform. The Methods section has been updated accordingly (Lines 36, 175, and 220).

Comments 2: Typographical Error in Results (Line 230). In the Results section, “SO and DO” appears to be a typographical error and should likely be corrected to “SO and DS.”Please re-check the entire manuscript for typographical or spelling inconsistencies to ensure that no errors remain.

Response 2:

Thank you for pointing out this typographical error. The term “SO and DO” has been corrected to “SO and DS” in the Results section (Line 230), and the entire manuscript has been rechecked to ensure no additional inconsistencies remain.

 Comments 3: The Limitations should be expanded to explicitly address the following points.

‘Wide heterogeneity in dose and duration among interventions’ and ‘The limited proportion of human studies relative to animal or cell-based studies’

Adding these points will more accurately reflect the constraints of the current evidence base and improve the scientific rigor of the review.

Response 3:

Thank you for this helpful suggestion. The Limitations section has been expanded to include the wide heterogeneity in dose and duration among interventions as well as the limited proportion of human studies relative to animal and cell-based research, and these revisions have been incorporated at Lines 528 and 553–558.